_Article_

# Viro3D: a comprehensive database of virus protein structure predictions

Ulad Litvin [ID] [1], Spyros Lytras [ID] [1,2], Alexander Jack[1], David L Robertson [ID] [1], Joseph Hughes [ID] [1,3] & Joe Grove [ID] [1,3 ✉]

## Abstract

**Viruses are genetic parasites of cellular life. Tolerance to genetic change, high mutation rates, adaptations to hosts, and immune escape have driven extensive sequence divergence of viral genes, hampering phylogenetic inference and functional annotation. Protein structure, however, is more conserved, allowing searches for distant homologs and revealing otherwise obscured evolutionary histories. Viruses are underrepresented in current protein structure databases, but this can be addressed by recent advances in machine learning. Using AlphaFold2-ColabFold and ESMFold, we predicted structures for >85,000 proteins from >4400 viruses, expanding viral coverage 30 times compared to experimental structures. Using this data, we map form and function across the human and animal virosphere and examine the evolutionary history of viral class-I fusion glycoproteins, revealing the potential origins of coronavirus spike glycoprotein. Our database, Viro3D (https:// viro3d.cvr.gla.ac.uk/), will allow the virology community to fully benefit from the structure prediction revolution, facilitating fundamental molecular virology and structure-informed design of therapies and vaccines.**

**Keywords** AlphaFold; ESMFold; Structural Bioinformatics; Virus Evolution; Viral Protein Structure
**Subject Categories** Computational Biology; Microbiology, Virology & Host Pathogen Interaction; Structural Biology

## Introduction

Viruses are obligate intracellular parasites whose replication depends on the metabolism and translational machinery of cellular organisms. Viruses have the capacity to evolve rapidly and infect organisms from all domains of life. They play crucial roles in ocean biogeochemical cycles (Breitbart et al, 2018; Suttle, 2007) and control of prokaryotic populations in the human gut microbiome (Shkoporov and Hill, 2019) but also infect and cause disease in crops, livestock, and humans. Virus particles are the most abundant

biological entities on our planet (Güemes et al, 2016), with metagenomic and metatranscriptomic studies only starting to sample the staggering genetic diversity of viral communities (Gregory et al, 2019; Hou et al, 2024; Zhang et al, 2019).

Genetic parasites, like viruses and mobile genetic elements, seem to be an inherent property of any replicating system (Iranzo et al, 2016b). Viruses likely emerged on multiple independent occasions, with the origin of the most ancient lineages probably coinciding with the origin of life and preceding the appearance of the last common ancestor of cellular organisms (LUCA) (Krupovic et al, 2019). Indeed, the genome of LUCA likely already carried an early antiviral defense system (Moody et al, 2024), implying that cellular organisms are inseparable from viruses and have been locked in a continuous arms race for the last 4 billion years. Independent evolutionary origins of viruses are reflected in the modern virus taxonomy by a specific taxonomic rank, the realms, which brings together viruses that share a set of conserved genes usually involved in genome replication or virion morphogenesis (Gorbalenya et al, 2020; Koonin et al, 2020).

The diversity of virus genome architectures with their frequent modular organization (Iranzo et al, 2016a), high mutation rates (Peck and Lauring, 2018), positive selection (Daugherty and Malik, 2012), and dependency on host cells, drive viruses to evolve faster than cellular organisms. Viruses exchange genetic material with their hosts and other viruses, turning their genomes into a mosaic of protein-coding and non-coding elements, with interweaved evolutionary histories (Mavrich and Hatfull, 2017). Countless examples of gene exchange between cellular organisms and viruses (Irwin et al, 2022) and co-opting of proteins that evolve to fulfil a new function (called exaptations) (Johnson, 2019; Koonin et al, 2022) highlight the evolutionary importance of genetic exchange between viruses and their hosts.

However, frequent genome reorganizations and high levels of divergence make the identification of gene function, investigation of deep phylogenetic relationships, and taxonomic assignments at higher ranks particularly challenging. In cases like these, comparison of protein structures for inference of evolutionary relatedness tends to be more reliable (Ravantti et al, 2020). Protein function is defined by tertiary structure, which is, as a result, more conserved than nucleotide or amino acid sequence (Ingles-Prieto et al, 2013; Chothia and Lesk, 1986; Illergård et al, 2009). Despite the striking diversity of viral sequences, viral protein structures are under-

[1]MRC-University of Glasgow Centre for Virus Research, Glasgow, UK. [2]Division of Systems Virology, Department of Microbiology and Immunology, The Institute of Medical Science, The University of Tokyo, Tokyo, Japan. [3]These authors contributed equally: Joseph Hughes, Joe Grove. ✉E-mail: Joe.Grove@glasgow.ac.uk

represented in public databases. Experimental protein structures from a viral source constitute less than 10% of the Protein Data Bank (PDB) (Berman et al, 2000).

Recent advances in machine learning have made it possible to predict protein structures from sequence alone, achieving accuracy similar to that of experimental structure determination (Akdel et al, 2022; Jumper et al, 2021; Lin et al, 2023). These state-of-the-art approaches have been applied at scale to produce comprehensive databases of predicted protein structures. The AlphaFold Structural Database (AFDB) (Varadi et al, 2024) contains more than 214 million models for proteins from UniProtKB (The UniProt Consortium, 2023) predicted using AlphaFold2 (Jumper et al, 2021). However, viral proteins were excluded from the prediction efforts. The Evolutionary Scale Modelling (ESM) Metagenomic Atlas became the largest structural database with more than 770 million models for proteins from metagenomic samples (MGnify database (Richardson et al, 2023)) predicted using ESMFold (Lin et al, 2023). Although this database contains some viral structures, they come predominantly from viruses of prokaryotes and unicellular eukaryotes.

Systematic exclusion of viral proteins from structure prediction efforts and their underrepresentation in public databases created a gap that is currently being addressed by the scientific community (Kim et al, 2025; Nomburg et al, 2024; Soh et al, 2024). In our study, we have generated 170,000 viral protein structure predictions from 4400 human and animal viruses using AlphaFold2-ColabFold (herein referred to as ColabFold) and ESMFold. We assessed the model quality produced by both methods and performed structural analysis of the proteins to expand their functional annotation. We also demonstrate that protein structure can guide the inference of deep phylogenetic relationships between viruses, using class-I membrane fusion glycoproteins as an example. To meet the needs of the virology community, we created Viro3D, a fully searchable and browsable database, allowing users to visualize and download proteome-level structural models for a virus of interest and explore similar structures present in other virus species (https://viro3d.cvr.gla.ac.uk/). We expect that this resource will find broad utility, accelerating fundamental molecular virology, enabling studies of virus evolution, and facilitating structure-informed development of therapies and vaccines.

## Results

### Systematic viral protein structure prediction with ColabFold and ESMFold

Our initial focus has been on predicting protein structures for human and animal viruses (Fig. 1A). We relied on data from the International Committee on Taxonomy of Viruses (ICTV) Virus Metadata Resource (VMR) (Lefkowitz et al, 2018; Data ref: ICTV VMR MSL38v2), which provides a comprehensive list of virus species and representative isolates, along with their GenBank accession numbers and host associations. At the time of our analysis, the latest version of the ICTV VMR included 3173 virus species infecting vertebrate and/or invertebrate hosts, represented by 4407 virus isolates/genotypes (see Dataset EV1). These viruses encoded a total of 71,274 proteins, spanning over 29.2 million amino acid residues (Fig. 1B). To simplify the analysis and protein structure prediction procedure, we excluded large polyproteins (≥2000 residues) from the dataset, replacing them with their

constituent matured cleaved proteins (annotated on GenBank as "mature peptides") and protein regions; this increased the total number of analyzed protein records to 85,162 (see Dataset EV2).

For protein structure prediction, we applied two state-of-the-art approaches: ColabFold (Mirdita et al, 2022), a method based on AlphaFold2 (Jumper et al, 2021) and dependent on multiple sequence alignments (MSAs), and ESMFold (Lin et al, 2023), a method that uses ESM-2, a transformer protein language model, and infers structure from input protein sequence alone. With ColabFold, we successfully predicted structures for all 85,162 records (27.2 million residues, covering 93.1% of total amino acid residues). Due to compute limitations in predicting longer proteins, ESMFold yielded slightly fewer predictions—84,964 protein records (27.0 million residues, covering 92.3% of amino acid residues). Structural coverage varied across viral realms, with most of them achieving coverage between 95% and 100% by both methods (Fig. EV1A). However, because of the lack of mature peptide and region annotation for many large polyproteins in the *Riboviria*, the coverage of this realm did not exceed 79.5% of total amino acid residues when we used ColabFold and 78.7% when we applied ESMFold. Nonetheless, since experimental structures in the PDB cover less than 3.3% of amino acid residues present in proteins of human and animal viruses (~890,000 amino acids), we have expanded the structural coverage for viral proteins by more than 30 times.

Overall, ColabFold models showed higher accuracy than those produced by ESMFold (Figs. 1C and EV1B). A total of 17.2 million residues predicted with ColabFold (63.3% of the modeled residues) were assigned high or very high quality, based on predicted local-distance difference test (pLDDT) score with the median pTM score of ColabFold predictions almost reaching 0.6 (Fig. EV1C). The number of high-quality residues increased to 87.4% (5.7 million residues) when evaluating models where a sequence homolog (defined as ≥30% identity) was available in the PDB at the time of AlphaFold2 training, dropping to 55.7% (11.5 million residues) for models without homologs (Fig. 1D). In contrast, only 31.6% of residues predicted by ESMFold (8.5 million residues) achieved high or very high quality with the mean pTM score of the models being around 0.3. Nonetheless, ESMFold models followed the same trend: with higher quality predictions for structures where a PDB sequence homolog was available at the time of training (47.6%, or 3.5 million residues) and lower quality for those without a homolog (25.5%, or 5 million residues). This suggests that training data may account for some higher accuracy models but, nonetheless, high-confidence predictions can still be achieved for sequences that were not well represented on the PDB (this may be particularly important for viral proteins, which are underrepresented in experimental structural data).

For almost 16% of the records, ESMFold models demonstrated higher pTM scores than ColabFold models (Fig. EV1D) with 9% of the models (7769 proteins) also having higher average pLDDT score (Fig. 1E). Although these improvements were typically minor, for 2% of the records (1753 proteins), the difference in pLDDT scores exceeded 10 points (Fig. 1F). One contributing factor to lower performance of ColabFold for these proteins was low MSA depth (Fig. EV1E), confirming the importance of having sufficient similar sequences available in public databases. The protein records where ESMFold outperformed ColabFold also tended to have longer sequences (Fig. EV1F).

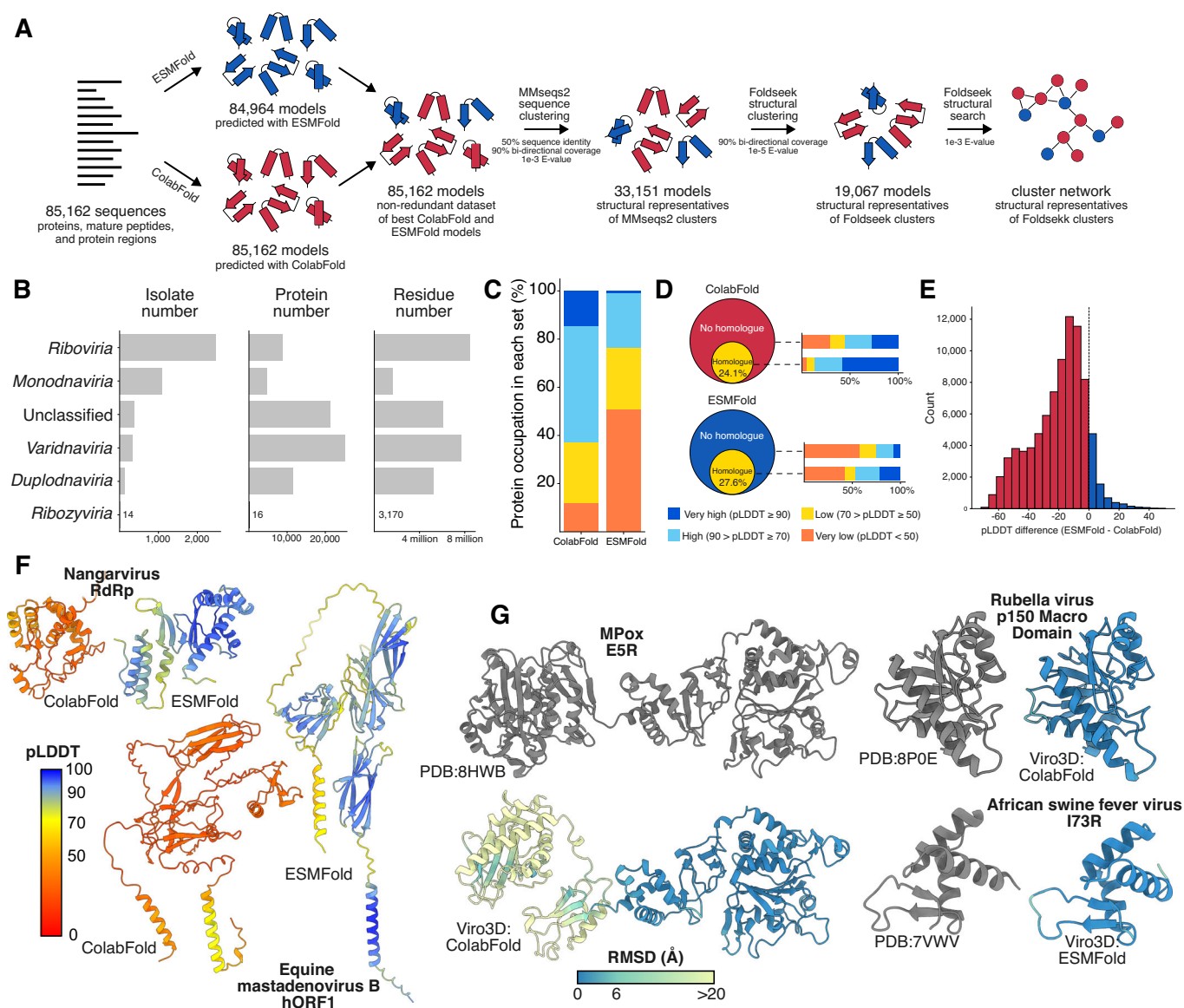

**Figure 1. Expanding structural coverage of the human and animal virosphere.**

(**A**) Summary of protein structure prediction, and downstream analyses (relevant to Fig. 2). We predicted structures for 85,162 proteins, peptides, and regions encoded by 4407 viruses from the ICTV VMR, with vertebrate and/or invertebrate hosts, using ColabFold and ESMFold. A non-redundant set of models was composed using models with the highest average pLDDT from either ColabFold or ESMFold. These were subjected to sequence and structure-based clustering, as described in the text. (**B**) Description of the target dataset with the number of virus isolates, proteins, and residues per viral realm. Unclassified viruses were those lacking realm annotation (e.g., *Baculoviridae*). (**C**) Percentage of ColabFold and ESMFold models with very high (dark blue), high (light blue), low (yellow), and very low (orange) average pLDDT score. (**D**) Difference in the confidence of predictions by ColabFold and ESMFold that are covered by PDB sequence homologs (defined as ≥30% sequence identity, yellow area) and those lacking PDB homologs (red and blue area, respectively). Plots display the proportion of total amino acid residues. (**E**) Distribution of differences in average pLDDT scores between ColabFold and ESMFold models for each protein record where both models are available. Positive pLDDT difference values indicate that ESMFold model is better (blue bars), negative value – ColabFold model is better (red bars). Comparison of pTM scores is provided in Fig. EV1. (**F**) Ribbon diagrams, color-coded by pLDDT, illustrating targets for which ESMFold produces a much more confident model. (**G**) Recent experimental structures (gray), determined after the AlphaFold2/ESMFold training data cutoff, compared to their cognate Viro3D predicted structures, which are color-coded by root-mean squared deviation (Å) after rigid-body alignment with their experimental counterparts. Low RSMD values (blue) indicate excellent agreement. Alignment of individual domains for Mpox E5R protein is provided in Fig. EV1G. Source data are available online for this figure.

We sought to test the accuracy of predicted models through comparison to experimental structures solved after the AlphaFold2 and ESMFold training dates, and for which there were no sequence homologs available on the PDB at the time of training. Very few targets met these criteria; nonetheless, we found excellent agreement between predicted and experimental structures (Fig. 1G), with the only major source of divergence being the positioning of, otherwise, well-folded domains (Fig. EV1G).

For all subsequent analyses, we created a non-redundant set of 85,162 models (Fig. 1A) representing each protein record with

either ColabFold or ESMFold model, depending on which one has the highest confidence (average pLDDT score).

In parallel with our work, two other efforts have recently performed systematic structure prediction for viral proteins. Nomburg et al predicted structures for 67,715 proteins from 4463 eukaryotic viruses derived from RefSeq entries in the NCBI Viruses portal (Nomburg et al, 2024). The Big Fantastic Virus Database (BFVD) did not consider viral taxonomy but used viral sequences from UniRef30 clusters to generate 351,242 protein structures (Kim et al, 2025). We used Foldseek (van Kempen et al, 2023) to compare Viro3D to these alternative resources (Appendix Fig. S1A). There was moderate overlap with Nomburg et al's dataset with 47,306 structures being unique to Viro3D; whilst only 12,894 BFVD structures were shared with Viro3D (72,268 being unique to Viro3D). To better understand the overlaps in data, we performed proteome-level comparisons for exemplar human pathogens. Here, the majority of entries had a match in the Nomburg et al dataset, but almost no proteins had a complete counterpart in BFVD, with many appearing only as truncated proteins (Appendix Fig. S1B). Indeed, comparison of protein lengths for all entries in BFVD and Viro3D, demonstrated that short sequences predominate in BFVD (Appendix Fig. S1C). This likely reflects the composition of the UniRef30 clustered sequences that underly BFVD (Kim et al, 2025). The high frequency of matches between Viro3D and the Nomburg et al dataset permitted side-by-side comparison of model quality, assessed by pLDDT prediction confidence. Here, Viro3D models outperformed their counterparts in Nomburg et al (Appendix Fig. S1D,E). This is explained by the MSA generation method (drawing only on RefSeq virus sequences) and structure prediction workflow of Nomburg et al (Nomburg et al, 2024). The benefits of Viro3D are well illustrated by comparisons of proteome structures from the PR8 isolate of influenza A virus (a commonly used model system for experimental virology), where Viro3D provides complete high-confidence models for all proteins (Appendix Fig. S1F). Therefore, whilst the Nomburg et al dataset covers a wider diversity of virology (including viruses of unicellular eukaryotes) and BFVD represents an excellent survey of structural diversity across the entirety of virology, Viro3D is the most comprehensive database of high-quality protein structure predictions for human and animal viruses.

## Exploration of the viral protein structure space by clustering and network analysis

We started with clustering the non-redundant set based on 90% bi-directional sequence overlap and 50% sequence identity that resulted in 33,151 sequence clusters. Then we selected a representative for each cluster, taking the highest confidence model (by average pLDDT score), and clustered this set of structural representatives based on 90% bi-directional structure overlap and 1e-5 e-value of the structural alignment (see "Methods"). This produced 19,067 structural clusters, 64.35% of which contained only one member (singleton clusters). Finally, we performed a Foldseek search among the representatives of these clusters to generate a structure similarity network of the viral proteins.

By applying restrictive clustering parameters, we ensured high structural homogeneity and consistency of functional annotation within each cluster (Appendix Fig. S2) but allowed homologous viral proteins to form multiple structural clusters. For instance, one of the most abundant proteins in our dataset, RNA-dependent RNA polymerase (RdRp), formed at least 110 structural clusters. The structure similarity network allowed us to address this issue by capturing communities of clusters that possess the same general protein fold or share a protein domain.

To demonstrate the power of structural network analysis, we examined the distribution of various common and hallmark viral proteins, many of which are used to define viral realms (Simmonds et al, 2023). This includes proteins involved in genome replication: RdRp, reverse transcriptase (RT), and DNA polymerase B (PolB); virion morphogenesis: single jelly roll (SJR), double jelly roll (DJR), and HK97 major capsid proteins (MCP); and membrane fusion: class-I, class-II, and class-III fusion glycoproteins (FG). We used Foldseek to identify communities of hallmark proteins by querying our structural network with a single experimental structure representative of each hallmark group (Fig. 2A). Protein communities located in the center of the network (e.g., RdRp, RT and SJR MCP) tend to be strongly interconnected, likely because of the clusters of polyproteins that possess multiple functional regions and therefore bring different communities together. This network-based structural search enabled the identification of significantly more proteins than comparable searches using either a standalone Foldseek structural search on a non-redundant set of 85,162 structures or hidden Markov model (HMM) profiles via HHblits (Remmert et al, 2012) (Fig. 2B). For example, it resulted in a 90% increase in identified RdRp structures—an improvement not achieved by HMM-profile searches, even after five iterations (Fig. EV2C). While HHblits identified distant RdRp homologs with 12.5% amino acid identity to the probe, the structural network approach extended this further, revealing homologs with as little as 6.5% identity (Fig. EV2B).

We plotted the distribution of hallmark proteins across viral families (Appendix Fig. S3) and realms (Fig. 2C). Here, the distribution of proteins correlates well with ICTV realm classification (Simmonds et al, 2023). For example, the HK97 capsid fold is a defining feature of the *Duplodnaviria* and, in our analysis, is confined to this realm (Fig. 2C). A similar relationship is seen for the double jelly roll capsid fold, which is a defining feature of the *Varidnaviria*.

RdRp or RT are hallmarks of the *Riboviria*, and whilst we detect these proteins in the majority of expected species, ~9% lacked Foldseek network hits for RdRp or RT. A detailed comparison to Pfam annotations (Fig. EV2) showed that most of the missing records were short regions and simply did not meet the requirement of 90% coverage with the RdRp reference structure (Fig. EV2E). Moreover, an additional barrier to identification of RdRps/RTs is partial genome coverage and incomplete gene annotation (Fig. EV2D); this serves as a further reminder that any effort in systematic structure prediction is limited by the quality, curation, and annotation of underlying sequence data. We also showed that five members of the *Varidnaviria* realm (*Poxviridae* and *Iridoviridae* families) possess an RT in addition to PolB. Four of these RTs are in a community of RdRp clusters (reflecting their shared ancestry) and therefore can be found using an RdRp probe, while one is clustered with other RTs.

PolB is abundant in the realms *Varidnaviria*, *Duplodnaviria*, and several viral families that currently do not belong to any realm (e.g., *Baculoviridae*). We also found a PolB in the *Bidnaviridae* family which belongs to the *Monodnaviria* realm. As expected, Single jelly roll MCP is prevalent in the realms *Riboviria*, *Monodnaviria*, and family *Baculoviridae*. All three classes of fusion

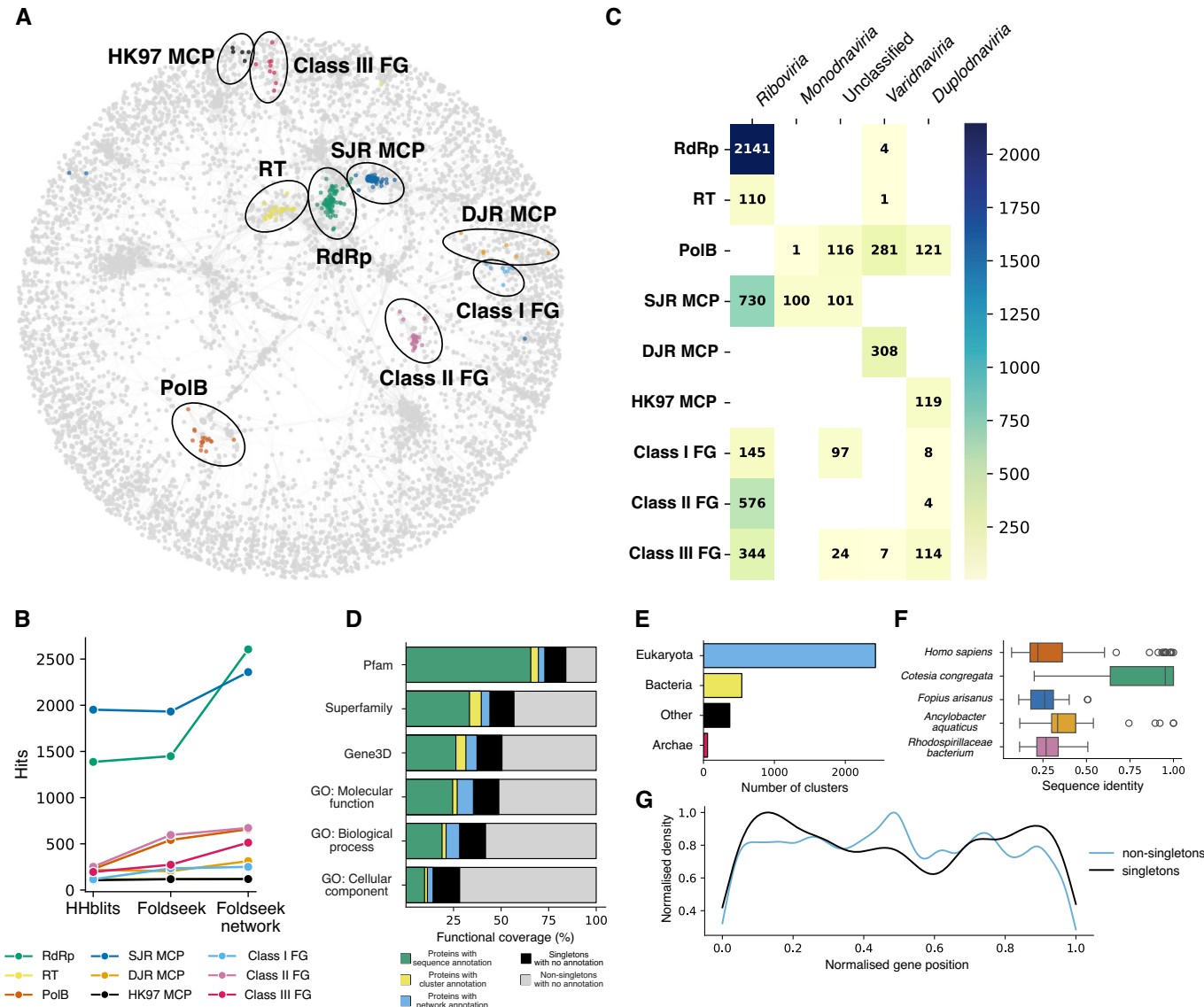

**Figure 2. Cluster and network analysis of viral protein structures.**

(A) Structure similarity network of viral proteins. Each node represents a cluster. Edges connect clusters that share structural similarity. Communities of hallmark proteins, which are used to define viral realms, are highlighted in different colors. RNA-dependent RNA polymerase (RdRp), reverse transcriptase (RT), DNA polymerase B (PolB), single jelly roll (SJR), double jelly roll (DJR) and HK97 major capsid proteins (MCP), class-I, class-II, and class-III fusion glycoproteins (FG). (B) Number of records identified for each hallmark protein reference using one iteration of HHblits search against Viro3D sequence dataset, or one iteration of Foldseek search against either Viro3D non-redundant structure dataset, or the network of structural clusters (see "Methods"). (C) Distribution of hallmark viral proteins across viral realms. Each square shows the number of viruses that possess structural homologs of hallmark proteins. (D) Expansion of function annotation using structural information. Each bar shows a percentage of protein records that possess Pfam annotation based on InterProScan (green), structural cluster expansion (yellow), and structural network expansion (light blue). The percentage of protein records that do not have a Pfam annotation are black (if they belong to singleton clusters) or light gray (if they belong to non-singleton cluster). (E) Number of viral protein clusters that have structural similarity to entries in the AFDB. Each bar represents the source of the AFDB structure. Structures coming from metagenomic, environmental, and unclassified samples were labeled as other. (F) Distribution of sequence identities between viral proteins and five cellular organisms from the AFDB with the highest number of structural matches. The bounds of the boxes represent the interquartile range (IQR, 25th–75th percentile), with the center line showing the median (50th percentile). Whiskers extend to the minima and maxima within 1.5×IQR. Individual data points outside the whiskers are outliers. *Homo sapiens* ($n = 142$), *Cotesia congregata* ($n = 69$), *Fopius arisanus* ($n = 38$), *Ancylobacter aquaticus* ($n = 34$), *Rhodospirillaceae bacterium* ($n = 33$). (G) Distribution of genome positions of viral protein-coding genes. Proteins that belong to singleton clusters are in black, protein that belong to non-singleton clusters are in light blue. Source data are available online for this figure.

glycoprotein are abundant in *Riboviria*, while class-III is the dominant glycoprotein in *Duplodnaviria*. Class-I and III fusion glycoproteins are also common among members of the *Baculoviridae* family (which is examined further, below).

Interestingly, PolB structural search also identified hits against other, non-PolB, proteins such as poxvirus F12L (AAL73754.1) and herpesvirus DNA helicase/primase (QBM10893.1) that possess shared domains or demonstrate similar fold architecture. Whilst

these "off-target" homologs are rare, this suggests there is a sensitivity/accuracy trade-off that could be balanced through changes in clustering and search parameters (e.g., higher or lower e-value thresholds). Nonetheless, by leveraging the fundamentally conserved nature of structure over sequence, combined with clustering and network analysis, we have achieved extremely sensitive detection of deep evolutionary relatedness, beyond even high-sensitivity sequence-based approaches. This permits efficient and accurate navigation of viral protein structure space and allows mapping of protein form and function across diverse species, as demonstrated by the consistent identification of hallmark proteins across viral realms.

We also used this approach to expand functional annotation, propagating sequence-based annotation using structural clusters and network. Out of 85,162 protein records, 65.6% have at least partial Pfam annotation (Fig. 2D). By propagating these annotations to unannotated cluster members, we expanded the functional coverage by 3.99% (3395 records). The propagation of annotations to clusters that do not have any annotated members using the structural network expanded the functional coverage by an additional 3.37% (2870 records, Fig. EV3). The consistency of propagated annotation is 100% for the majority (70.3%) of protein records (4403 out of 6265 records, Appendix Fig. S4). We also expanded Gene3D (Lewis et al, 2018) and Superfamily (Pandurangan et al, 2019) annotation, which gave an increase of 11.1% and 10.7%, respectively. Expansion of gene ontology (GO) annotation (The Gene Ontology Consortium, 2019) for molecular function, biological process, and cellular component increased the number of annotated records by 10.9%, 9.2% and 4.4%, respectively. Therefore, clustering of structures and network analysis permits the identification and/or functional annotation of as-yet unclassified viral proteins. Nonetheless, it is important to note that despite propagation, there is still a high proportion of proteins with no known function (Fig. 2D).

To estimate the number of protein structures shared between viruses and cellular organisms, we performed a structure similarity search between cluster representatives and the AlphaFold Structural Database (AFDB, Fig. 2E). Consistent with the notion that viruses are a source of novel protein folds (Nomburg et al, 2024), the majority of viral protein clusters do not share detectable homology with cellular life, with only 17.8% of clusters (3393 out of 19,067) having significant structural similarity to proteins in the AFDB. Of these homologs, 71.5% are coming from Eukaryota, 15.9% from Bacteria, 1.7% from Archaea, and 10.9% from metagenomic and environmental samples. However, this number is likely to be an overestimation because in many instances, hits are coming from endogenous or symbiotic viruses. For instance, many *Homo sapiens* hits with high sequence identity are proteins from the integrated *Human betaherpesvirus 6A* (Fig. 2F), while hits from *Cotesia congregata* and *Fopius artisans*, two species of parasitoid wasps, are proteins from symbiotic viruses of genera *Bracoviriform* and *Alphanudivirus*, respectively.

In all, 14.4% of the protein records form singleton clusters and potentially represent structural novelty. Our species-focussed approach captured the genomic context of all predicted structures and, therefore, allowed us to investigate the genome positions of singleton and non-singleton clusters (Fig. 2G). Interestingly, unique protein structures are more frequently found at the start or end of linear viral genomes, while common protein structures (which are likely to be associated with critical functions) tend to have a more uniform distribution across the genome length. The picture is different for some viral realms, some of which have predominantly circular genomes (e.g., Monodnaviria; Appendix Fig. S5), nonetheless, this suggests that genomic termini are hotspots for evolutionary innovation, which may drive the emergence of novel protein functions and adaptations to hosts.

## The deep evolutionary history of class-I fusion glycoproteins

Class-I fusion glycoproteins can be found in a wide range of important human pathogens, including SARS-CoV-2, HIV, Influenza, and Ebola, and have been exapted by mammals to mediate placental morphogenesis (Mi et al, 2000). Mechanistic knowledge of these proteins informs rational vaccine design (Sanders and Moore, 2021). It is thought that class-I fusion glycoproteins share a common origin, however, extensive sequence divergence has all but erased this deep ancestry at the amino acid level. Therefore, classification of fusion glycoproteins is typically achieved through structural and functional characterization. Indeed, recent experimental structures of retroviral Env class-I glycoproteins suggest shared ancestry with fusion glycoproteins of negative-sense RNA viruses (Calcraft et al, 2024; Fernández et al, 2024); this evolutionary relationship is also supported by analysis of the viral "fossil-record" provided by endogenous viral elements (Nino Barreat and Katzourakis, 2024). Nonetheless, traditional structural biology and/or sequence analyses provide only a fragmentary picture of glycoprotein ancestry. We reasoned that homology detection, achieved through structural similarity and network analysis (Fig. 2A,C), would permit a virosphere-wide survey of class-I fusion glycoproteins and provide a clear view of their deep evolutionary history.

First, to gain a global perspective, we generated a structure-informed map of the human and animal virosphere. This allows simultaneous visualization of all viruses represented in Viro3D, onto which the distribution of proteins can be projected. This was achieved by systematic structural comparison of each virus' proteome (see "Methods"), resulting in a scatter plot, with viruses segregated by structural similarity (Fig. 3A), which broadly recapitulates viral taxonomy (Appendix Fig. S6).

Mapping the viruses that possess class-I fusion glycoproteins (identified by structural homology to RSV-F, Fig. 2C) demonstrates a distribution across viral realms, indicative of extensive genetic exchange (Fig. 3B). Alongside expected instances of class-I fusion glycoproteins in negative-sense RNA viruses, reverse-transcribing viruses, and positive-sense Nidoviruses, we discovered previously unknown class-I fusion glycoproteins in the *Herpesvirales* and *Baculoviridae*. Note that detection was achieved using a single structural reference (RSV F), we expect further instances may be found (e.g., in the *Retroviridae*) using alternative reference structures.

The presence of class-I fusion glycoproteins in the *Herpesvirales* and *Baculoviridae* was not expected, as both of these viral taxa are more commonly known to possess class-III fusogens; gB in *Herpesvirales* (Heldwein et al, 2006) and gp64 in *Baculoviridae* (Kadlec et al, 2008). To examine this further, we surveyed the proteomes of all *Herpesvirales* and *Baculoviridae* species represented on Viro3D, using Foldseek to search for structural homology

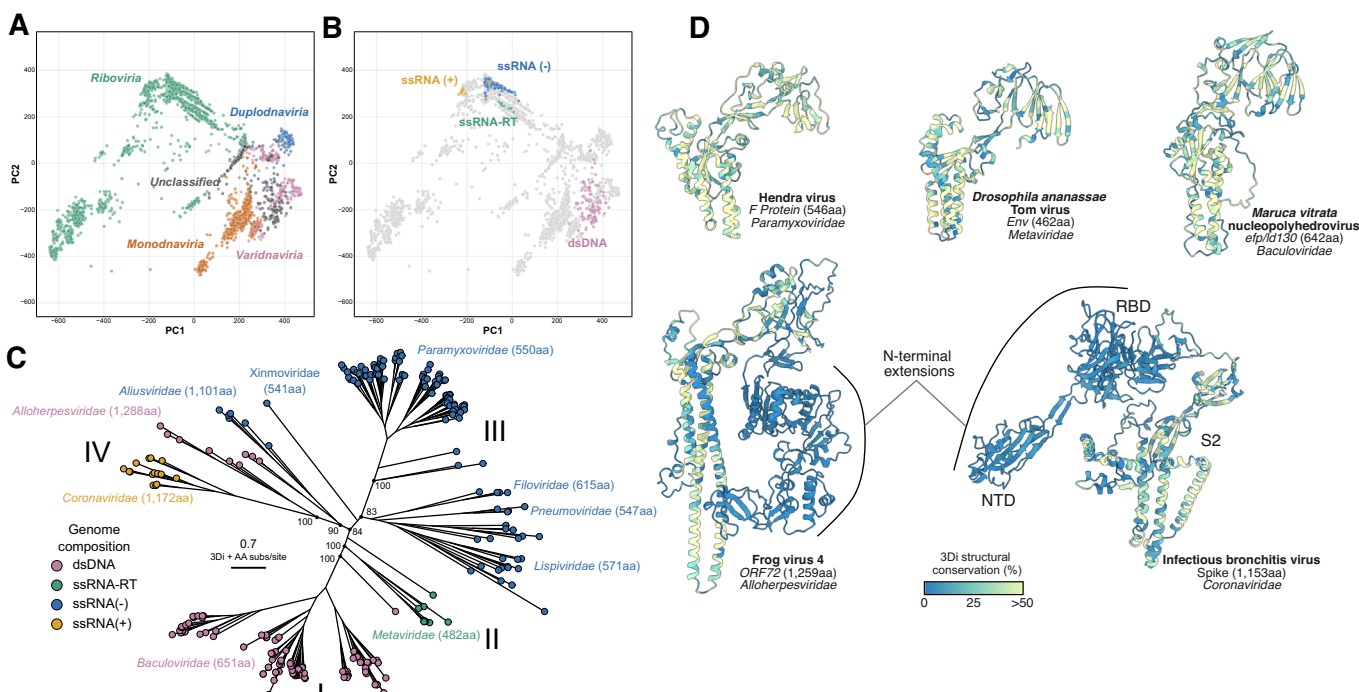

**Figure 3. The deep evolutionary history of class-I fusion glycoproteins.**

(A) Structure-informed map of the human and animal virosphere, each data point represents a virus in the Viro3D database (see Methods). Viruses are color-coded by realm. (B) Distribution of class-I fusion proteins across the human and animal virosphere. Colored data points represent viruses in which we detect a class-I fusion protein using structural searches and network analysis, with RSV-F as a reference. Viruses are color-coded by genome composition, as labeled. (C) Combined 3Di and amino acid phylogenetic inference of identified class-I fusion proteins, achieved by structure-guided sequence alignment using Famsa3di. Tips and labels are colored by genome composition. Labels indicate viral genera and, in parentheses, the corresponding average length of glycoprotein. Major clades are labeled, as described in the main text. Node support values are provided at important branch points. Scale bar represents 3Di and amino acid substitutions per site. (D) Example structures: ribbon diagrams are color-coded by structural conservation, assessed by the 3Di alignment, as denoted in the key. Labels indicate N-terminal extensions, present in clade IV glycoproteins, and pertinent regions of spike (N-terminal domain: NTD, receptor-binding domain: RBD, and the S2 subunit). Note, predicted structures are monomeric, as opposed to the native trimeric state of class-I fusion glycoproteins, and domain positioning (e.g., NTD and RBD) may be inaccurate. Source data are available online for this figure.

to a range of experimentally determined class-I and class-III fusion protein structures. Mapping this information against DNA polymerase phylogeny reveals the distribution of fusion mechanisms within either taxonomic group (Fig. EV4). The vast majority of viruses within the *Herpesvirales* exhibit strong Foldseek hits against class-III references (particularly vesicular stomatitis virus G protein and human herpesvirus-5 gB). However, a minor, highly divergent, clade containing aquatic herpesviruses (e.g., Channel catfish virus), possess no class-III homologs, instead exhibiting Foldseek hits against class-I proteins (particularly the SARS-CoV-2 spike). Class-I glycoproteins predominate in the *Baculoviridae*, with only a subset possessing class-III fusion glycoproteins; consistent with the confinement of gp64 to group I alphabaculoviruses (and a single species of betabaculoviruses (Ardisson-Araújo et al, 2016)).

To gain further insights into the evolutionary history of the class-I fusion mechanism, we performed structure-guided sequence alignment, permitting phylogenetic inference using both structure and sequence information (Fig. 3C; Appendix Fig. S7). This reveals four major clades (arbitrarily designated I-IV). The first clade is comprised of proteins from the *Baculoviridae*. The second contains reverse-transcribing viruses, from the *Metaviridae* (a family of LTR retrotransposons (Llorens et al, 2020)), which share insect hosts with the *Baculoviridae*, providing a feasible route of genetic

exchange (Ozers and Friesen, 1996). The third clade is monophyletic for the *Mononegavirales*, this may suggest a single genetic acquisition by an ancestral negative-sense RNA virus, from which class-I fusion glycoproteins propagated throughout the order. The final clade (IV) contains a mixture of viral taxonomies, including negative-sense *Aliusviridae*, positive-sense *Coronaviridae*, and dsDNA *Herpesvirales*. These clade IV glycoproteins are particularly long when compared to the rest of the phylogeny (most being >1100 residues; Fig. 3C), suggesting adaptive diversification.

Comparison of structures from across the phylogeny reveals a central structurally conserved architecture common to all of the identified class-I fusion glycoproteins (Fig. 3D), consistent with recent experimental studies (Calcraft et al, 2024; Fernández et al, 2024). Representative clade IV glycoproteins from an aquatic herpesvirus and *Coronaviridae* (infectious bronchitis virus), however, possess very long N-terminal extensions; this includes the N-terminal domain (NTD) and receptor-binding domain (RBD) that constitute the majority of the S1 subunit in coronavirus spike. Moreover, the region conserved across all species broadly corresponds to the fusogenic S2 subunit of spike. Notably, the *Coronaviridae* form a high-confidence branch with the *Alloherpesviridae* (Fig. 3C). This is consistent with the Foldseek homology for SARS-CoV-2 spike apparent in the aquatic herpesviruses (Fig. EV4).

This suggests that the coronavirus spike glycoprotein may have originated from genetic exchange with an ancestral herpesvirus.

## Discussion

In this work, we expanded the structural coverage for viral proteins by 30 times compared to that of experimental structures by generating structural models for 85,000 proteins from 4400 human and animal viruses. Approximately 64% of the produced protein models are confident predictions with an average pLDDT score above 70. Moreover, 65% of residues in the predicted dataset have high or very high confidence. We confirmed that, in general, ColabFold has a higher chance of producing a confident structure than ESMFold. However, since ESMFold models have higher confidence for almost 10% of protein records, do not require an MSA, and are usually much faster to generate, the combination of both approaches is highly beneficial.

Parallel efforts to ours have produced alternative repositories of viral structure predictions. While the Nomburg et al dataset covers a greater diversity of eukaryotic viruses, and BFVD surveys a wider diversity of viral sequence and structure space, we demonstrated that Viro3D contains the highest-quality and most complete structural models for human and animal viruses. We anticipate that Viro3D will be particularly valuable for informing experimental molecular virology and evolutionary studies of functional diversity across the animal virosphere.

By performing clustering of protein structures encoded by human and animal viruses, the diversity of viral proteins was reduced to ~19,000 distinct protein structures; almost 65% of these structures are unique within our dataset. Interestingly, in viruses with linear genomes (realms *Varidnaviria*, *Duplodnaviria*, and, to some extent, *Riboviria*), these unique proteins are usually found closer to the ends of the genomes, which probably represent hot spots of gene acquisition or de novo gene origination.

More than 82% of distinct viral structures are unique to viruses and do not have apparent structural homologs in cellular organisms. This high percentage of unique viral structures may seem surprising given the extensive horizontal gene transfer between viruses and their hosts, and may suggest that even acquired proteins undergo extensive remodeling and diversification in the context of a viral genome. Therefore, we acknowledge that the number of viral proteins retaining more distant structural similarity to cellular components may be higher than that demonstrated in our analysis; indeed, parallel studies have identified slightly higher numbers of cellular homologs in viruses (Nomburg et al, 2024).

We showed that structure similarity searches combined with a structural network allowed us to substantially accelerate the search process by increasing the number of homologous structures identified in a single run. For instance, using single RdRp and RT structures, we identified RNA-directed polymerases for 90% of viruses in the *Riboviria* realm. This demonstrates that structural searches and network analysis are an extremely efficient means of traversing large evolutionary distances.

Classification of viral realms (the highest taxonomic rank for viruses) is achieved through the identification of hallmark genes. These are, semi-arbitrarily chosen, proteins involved in genome replication and capsid morphogenesis. For example, RdRp and RT are hallmarks of the *Riboviria*, while DJR-MCP and HK97 MCP are specific to *Varidnaviria* and *Duplodnaviria*, respectively (Krupovic

and Koonin, 2017). PolB is distributed more broadly: realms *Varidnaviria*, *Duplodnaviria*, and some unclassified DNA viruses (e.g., *Baculoviridae*). Our analyses are largely consistent with, and verify, these taxonomic relationships, with rare exceptions. Most families in the *Monodnaviria* rely on rolling-circle replication endonuclease (Kazlauskas et al, 2019), however, we found PolB in one family of this realm: *Bidnaviridae* (Krupovic and Koonin, 2014). Similarly, we found multiple instances of RT/RdRP in members of the *Varidnaviria* (*Poxviridae* and *Iridoviridae*). These edge cases likely reflect inter-realm horizontal gene transfer and serve as reminders that the mosaic nature of viral genomes will often present challenges for taxonomic classification. As an additional example, membrane fusion glycoproteins are common but not considered hallmark proteins, and are spread across multiple realms. Class-III fusion glycoproteins seem to be the most widely distributed among human and animal viruses, being absent only in the realms *Monodnaviria* and *Ribozyviria*.

Fusion glycoproteins are fundamentally important for virus transmission, pathogenesis, and spillover, and are major targets for host immunity. Viro3D has provided a new opportunity to comprehensively survey the distribution of glycoproteins and, using structure-guided approaches, infer their evolutionary past. Using just a single structural reference to query Viro3D (RSV F glycoprotein), we identified 251 highly diverse class-I fusion glycoproteins, many sharing <10% sequence identity. This includes both known instances (e.g., in *Mononegavirales*) and unexpected hits within the *Herpesvirales* and *Baculoviridae*. Systematic Foldseek surveys across herpes- and baculoviruses taxa suggest a mixture of both class-III and class-I fusion mechanisms. In fact, class-I fusion appears to predominate in the *Baculoviridae* (including in those species with gp64) and homology frequently maps to known/suspected fusogens such as the F protein of betabaculoviruses (Rohrmann, 2019). This suggests that class-I glycoproteins are the ancestral fusion mechanism in Baculoviruses, with the class-III gp64 being acquired as an alternative virus entry system (Rohrmann, 2019). Indeed, gp64 may have originated through a horizontal gene transfer from thogotoviruses (Milhomem Pilati Rodrigues et al, 2025).

Combined structure and sequence-based phylogenetics provided the first view of the deep evolutionary history of class-I fusion glycoproteins. This is consistent with multiple independent horizontal gene transfers from an, as yet unidentified, ancestral source. Whilst all of the identified glycoproteins share a structurally conserved central fold, one clade in particular (including coronaviruses and aquatic herpesviruses) shows signs of extensive adaptive diversification. These glycoproteins have doubled in size by gaining novel regulatory elements, such as the NTD and RBD in the S1 subunit of spike. This provides an evolutionary perspective on membrane fusion in coronaviruses: the S1 subunit is proteolytically separated from spike and is shed following receptor engagement, leaving the structurally conserved S2 to mediate fusion (Grunst et al, 2024). Thus, to achieve cellular entry, coronaviruses undergo a regulated unmasking of an ancestral fusogen, which is fundamentally conserved across all class-I fusion glycoproteins. The phylogenetic topology of this clade suggests that coronaviruses gained their spike glycoprotein by genetic exchange with an ancestor of the known aquatic herpesviruses; albeit, we cannot eliminate the involvement of other unsampled, or extinct, taxa in this exchange. In summary, using class-I fusion glycoproteins as an exemplar, we have demonstrated that structure-guided discovery, enabled by Viro3D, is likely to provide unprecedented insights on the origins and evolution of viruses and their defining proteins.

Whether for lab-based investigators wishing to bring three-dimensional context to their molecular virology experiments, researchers performing structure-guided design of therapies, or computational biologists studying deep virus evolution, we expect the rich structural dataset presented here to be a valuable resource for the virology community. Viro3D is fully searchable and browsable here: https://viro3d.cvr.gla.ac.uk/.

# Methods

## Reagents and tools table

| Reagent/resource | Reference or source | Identifier or catalog number |
|---|---|---|
| **Experimental models** | | |
| None | | |
| **Recombinant DNA** | | |
| None | | |
| **Antibodies** | | |
| None | | |
| **Oligonucleotides and other sequence-based reagents** | | |
| None | | |
| **Chemicals, enzymes, and other reagents** | | |
| None | | |
| **Software** | | |
| LocalColabFold v1.5.2 | https://github.com/YoshitakaMo/localcolabfold | |
| ESM-2 v1.0.3 | https://github.com/facebookresearch/esm | |
| MMseqs2 v15.6f452 | https://github.com/soedinglab/MMseqs2 | |
| Foldseek v9.427df8a | https://github.com/steineggerlab/foldseek | |
| HHsuite v3.3.0 | https://github.com/soedinglab/hh-suite | |
| InterProScan v5.69-101.0 | https://github.com/ebi-pf-team/interproscan | |
| FoldMason v99fbda50e6c296c9fdf15d05fefedbb91f4efe84 | https://github.com/steineggerlab/foldmason | |
| FAMSA3di | https://doi.org/10.1101/2023.12.12.571181 | |
| TrimAI v1.5.0 | https://github.com/inab/trimal | |
| IQ-TREE v2.3.6 | https://github.com/iqtree/iqtree2 | |
| UCSF ChimeraX v1.8 | https://www.cgl.ucsf.edu/chimerax/ | |
| Python v3.12.0 | https://www.python.org/ | |
| NetworkX v3.3 | https://networkx.org/ | |
| Pandas v2.1.3 | https://pandas.pydata.org/ | |
| NumPy v1.26.2 | https://numpy.org/ | |
| Pfam2go v1.1.2 | https://pypi.org/project/pfam2go/ | |
| Matplotlib-base v3.8.2 | https://matplotlib.org/ | |
| Matplotlib-venn v1.1.1 | https://matplotlib.org/ | |
| Seaborn v0.13.0 | https://seaborn.pydata.org/ | |
| Plotly v5.19.0 | https://plotly.com/ | |
| **Other** | | |
| colabfold_envdb_202108 database | https://colabfold.mmseqs.com/ | |
| pdb100_230517 database | https://colabfold.mmseqs.com/ | |
| uniref30_2302 database | https://colabfold.mmseqs.com/ | |

## Dataset preparation

In total, 6721 GenBank nucleotide accession numbers of virus genomes or genome segments were retrieved for 4407 virus isolates (corresponding to 3106 virus species, Dataset EV1) with invertebrates and/or vertebrates host annotation from the ICTV Virus Metadata Resource (Lefkowitz et al, 2018; Data ref: ICTV VMR MSL38v2). From the viral genomes, we extracted 71,269 protein records encoded by viral protein-coding genes annotation on GenBank. Since many viral genomes encoded polyproteins, we also relied on GenBank annotation to retrieve 4070 mature peptides from 489 records annotated as polyproteins and 11,786 protein regions from 2087 proteins with length greater or equal to 2000 amino acids (aa); these annotated sequences frequently relate to the mature proteins that are proteolytically liberated from the polyprotein. In all, 360 proteins had both peptide and region annotation. All records had protein sequence lengths greater than 10 aa. In total, 69,053 proteins without peptide or region annotation, 11,786 protein regions, 4070 mature peptides, and 253 shortest polyproteins with peptide and/ or region annotation were used for protein structure prediction (Dataset EV2).

## Protein structure prediction with ColabFold

Protein models for all 85,162 records were predicted using LocalColabFold v.1.5.2 with default settings (Mirdita et al, 2022). Multiple sequence alignments (MSAs) were constructed for each record using MMseqs2 v.15.6f452 (Steinegger and Söding, 2017), and colabfold_envdb_202108, pdb100_230517, uniref30_2302 databases installed locally. For each protein record, ColabFold produced five models using three recycles and no PDB templates. All models were ranked based on the mean predicted local-distance difference test (pLDDT) score. The top-ranked model was subjected to constrained relaxation by gradient descent in the Amber force field (Pearlman et al, 1995; Hornak et al, 2006; Mirdita et al, 2022) with the following settings: max_iterations 2000; tolerance 2.39; stiffness 10.0. The relaxed models were used for structural analysis.

## Protein structure prediction with ESMFold

We also predicted 84,964 protein models for 69,043 proteins without peptide or region annotation, 4070 mature peptides, 11,767 protein regions, and 84 proteins with peptide and/or region annotation using ESMFold module of ESM-2 v.1.0.3

(Lin et al, 2023) with default settings. For each record ESMFold produced 1 model using four recycles. For records with protein length greater than 1236 aa, the following settings were used: --max-tokens-per-batch 1 --chunk-size 128. Due to memory constraints on the GPU, we were unable to predict models for records with protein length greater than 2840 aa. ESMFold models were subjected to relaxation in the Amber force field as explained above.

## Estimation of structural coverage expansion

To estimate the expansion of structural coverage relative to experimentally determined structures available in the PDB (Berman et al, 2000), and to consider the influence of PDB training data on prediction accuracy, we performed a sequence similarity search of 71,269 protein records against the PDB 2024-02-20 using MMseqs2 v.15.6f452 (Steinegger and Söding, 2017) easy-search command with the following sensitivity settings: -s 7.5 --max-seqs 100,000 -e 1e-3. To calculate the percentage of residues covered by PDB structures, only PDB hits with sequence identity greater or equal to 95% were retained. To calculate the percentage of residues covered by homologs from the PDB, PDB entries released after 2018-04-30 or 2020-05-01, training data cut-offs for AlphaFold2 (Jumper et al, 2021) and for ESMFold (Lin et al, 2023), respectively, were filtered out, only PDB hits with sequence identity of 30% or greater were retained.

## Clustering procedure

Overall, 79,036 ColabFold and 6126 ESMFold models with the highest pLDDT score per protein record were combined into a non-redundant dataset for the downstream structural analysis. ESMFold models were used as representatives of protein records if they had mean pLDDT score greater than 50, and this score was greater than the mean pLDDT score of a corresponding ColabFold structure. First, we clustered 85,162 protein records based on their sequence similarity using MMseqs2 v.15.6f452 (Steinegger and Söding, 2017) easy-cluster command with minimum sequence identity of 50% and bi-directional coverage of 90% (--min-seq-id 0.5 --cov-mode 0 -c 0.9 -e 1e-3 --cluster-mode 0). Second, we defined structural representatives for 33,151 MMseqs2 clusters by choosing the member with the highest model pLDDT score. We clustered 33,151 structure representatives based on structural similarity using Foldseek v.9.427df8a (van Kempen et al, 2023) easy-cluster command with no minimum sequence identity but bi-directional coverage of 90% (--min-seq-id 0 --cov-mode 0 -c 0.9 -e 1e-5), producing 19,067 structural clusters (Dataset EV3). Foldseek e-value cutoff (1e-5) is an arbitrary threshold chosen to achieve high-sensitivity for detecting structural homologs; however, this can result in some "off-target" hits such as the herpesvirus DNA helicase/primase proteins identified by homology to DNA polymerase B.

## Structural and functional homogeneity of clusters

To estimate structural homogeneity within each non-singleton cluster, we extracted LDDT and alignment TM scores from a reciprocal Foldseek v.9.427df8a (van Kempen et al, 2023) easy-search of the non-redundant dataset with next settings: --exhaustive-search -e 0.1. For each cluster, we calculated the median LDDT and TM score of the alignments between the cluster representative

and other cluster members. To estimate the consistency of function annotation within non-singleton clusters, we relied on a sequence-based Pfam annotation acquired using InterProScan v.5.69-101.0 (Jones et al, 2014) with default settings. For each protein record, we retained a functional annotation with the highest sequence coverage and calculated the percentage of annotated records that share the predominant Pfam annotation for each cluster containing at least two annotated members.

## Structure similarity network analysis

Similar to MMseqs2 clusters, we defined structural representatives for 19,067 Foldseek clusters by choosing the model with the highest pLDDT score from each cluster. We performed a structural comparison of the representatives using Foldseek v.9.427df8a (van Kempen et al, 2023) easy-search command with default settings and applied the results to construct a structure similarity network using NetworkX v.3.3 (Hagberg et al, 2008). Only structural representatives with pLDDT score above 50 and at least one connection with Foldseek e-value below 1e-3 were retained, producing a graph with 7812 nodes and 37,751 edges (Dataset EV4). The graph was visualized using NetworkX spring_layout() function with k of 0.3 and 500 iterations.

## Identification of hallmark proteins and fusion glycoproteins

We started with one iteration of Foldseek v.9.427df8a (van Kempen et al, 2023) easy-search against the non-redundant structural dataset with the following settings -e 1e-5 --max-seqs 10,000 and using a set of experimentally determined structures as probes: poliovirus RNA-dependent RNA polymerase (PDB ID: 4R0E (Moustafa et al, 2014)), human immunodeficiency virus reverse transcriptase (1HMV (Rodgers et al, 1995)), Phi29 DNA polymerase B (2PY5 (Berman et al, 2007)), circovirus rolling-circle replication endonuclease (8H56 (Guan et al, 2023)), papillomavirus hexameric superfamily 3 helicase (5A9K (Chaban et al, 2015)), poliovirus VP3 protein (8E8R (Charnesky et al, 2023)) as single jelly roll capsid protein, human adenovirus 5 hexon protein (6B1T (Dai et al, 2017)) as double jelly roll capsid protein, varicella zoster virus HK97 capsid protein (6LGL (Wang et al, 2020)), respiratory syncytial virus F protein (6APB (Goodwin et al, 2018)) as a class-I fusion glycoprotein, spondweni virus E protein (6ZQI (Renner et al, 2021)) as a class-II fusion glycoprotein, human cytomegalovirus gB protein (7KDP (Liu et al, 2021)) as a class-III fusion glycoprotein. To test if querying the structure similarity network increases the number of hits, we performed a separate Foldseek easy-search using the same list of probes and search parameters, but this time only against a set of 19,067 structural representatives of Foldseek clusters. We then expanded the number of structurally similar clusters by adding Foldseek clusters that have a significant network connection (e-value 1e-5 and 90% coverage of the Foldseek hit). The final list of the hits contained all members of the Foldseek clusters identified using this approach. For comparison we also performed up to five iterations of a profile-profile HHsuite v3.3.0 HHblits search (Remmert et al, 2012) against protein sequences in the non-redundant dataset using sequences from the hallmark PDBs as probes with following settings -e 1e-3 -Z 5000 -B 5000.

## Expansion of functional annotations

To demonstrate how structural information can be used to expand functional annotation, first we performed a sequence-based annotation of 85,162 proteins using InterProScan v.5.69-101.0 (Jones et al, 2014) with default settings. We focused on annotations from Pfam (Mistry et al, 2021), Superfamily (Pandurangan et al, 2019), and Gene3D (Lewis et al, 2018) databases because they provided the highest number of annotated proteins: 55,869, 28,431, and 22,336 proteins, respectively. From each database, we retained only one functional annotation for each protein based on the sequence coverage; shorter annotations were neglected in this analysis. To annotate proteins with unknown function within annotated clusters, we used structural clusters: for each cluster, we calculated frequencies of annotations coming from annotated members and propagated the predominant annotation to members with unknown function. To annotate clusters with unknown function, we used a structural network: we calculated frequencies of annotations coming from structural representatives of connected clusters and propagated the predominant annotation to the clusters with no annotation. In case of annotation expansion using the structural network, only connections with Foldseek e-value below 1e-3 and 90% coverage of InterProScan annotation were considered. Expansion of Gene Ontology (GO) terms (The Gene Ontology Consortium, 2019) was done in the same way based on GO IDs associated with Pfam annotations described above. A list of proteins with expanded Pfam annotation is available as Dataset EV5; the rest of the tables are available at https://doi.org/10.5281/zenodo.15622906.

## Structural comparison to the AlphaFold structural database

We compared 19,067 structural representatives of Foldseek clusters to protein models in the AlphaFold Structural Database (Varadi et al, 2024) using Foldseek v.9.427df8a (van Kempen et al, 2023) easy-search option with e-value of 1e-3. For each structural representative, we retained the hit with the lowest e-value.

## Comparison to alternative repositories of viral protein structure predictions

We systematically compared Viro3D to the dataset generated by Nomburg et al and to the Big Fantastic Virus Database (BFVD) (Kim et al, 2025; Nomburg et al, 2024). We first used Foldseek searches to assess overlap with Viro3D. Here, protein models were considered shared if they had a Viro3D Foldseek hit with an e-value lower than 1e-5, sequence identity equal or greater than 95% and query coverage equal or greater than 95%. Next, we examined structure predictions from a panel of priority human viral pathogens. Using taxonomic identifiers to find the relevant models, we compiled structure sets for each pathogen from Viro3D, Nomburg et al and BFVD. Using protein sequence to identify matching models (95% sequence similarity), we evaluated coverage of proteins (relative to Viro3D) and the confidence of structure prediction (assessed using pLDDT values).

## Building a structural-similarity map displaying all viral species

We first performed all-vs-all Foldseek comparison of the entire non-redundant structure set (85,162 models), therefore surveying global structural-similarity. Using this, for any given virus' proteome (i.e., collection of structures), we extracted the lowest e-value against each of the 85,162 models. If no e-value was present for a given virus-model pair, we substituted an arbitrary high value of 10. This resulted in a matrix where each of the 4407 viruses is represented by 85,162 structural similarity scores. We then used principal component dimensionality reduction to group viruses based on the structural similarity of their respective proteomes.

## Phylogenetic analysis of the class I fusion glycoprotein

Class-I fusion proteins were identified using Foldseek homology search (see above, identification of hallmark proteins and fusion glycoproteins), resulting in 259 structural models. These were filtered to remove those with extremely low confidence (pLDDT ≤40), resulting in a final set of 251 structures with mean pLDDT=70.4( +/−7.3) and mean pTM=0.58( +/− 0.08); a list of these proteins is provided in Dataset EV6. We note that confidence metric filtering may be important to remove erroneous homologs. These structures were aligned using two approaches: (i) FoldMason (Gilchrist et al, 2024) (easy-msa –report-mode 1) and (ii) the famsa3di method described in Puente-Lelievre et al (2024) (Deorowicz et al, 2016; Puente-Lelievre et al, 2024), producing corresponding 3Di and amino acid alignments. Subsequently, the phylogenetic relationships were determined using IQ-TREE v.2.3.6 (Minh et al, 2020) using the two 3Di alignments (FoldMason, famsa3di) (Appendix Fig. S7). Suitable substitution models were tested by BIC using ModelFinder (Kalyaanamoorthy et al, 2017) and the custom 3DI substitution matrix (Puente-Lelievre et al, 2024) with empirically counted frequencies from the alignment and accounting for rate heterogeneity under a FreeRate model was the best (for both 3Di alignments (five FreeRate categories for famsa3di, $3DI + F + R5$, and six categories for FoldMason, $3DI + F + R6$). To inform phylogenetic reconstruction with amino acid homology, we also performed partitioned model inferences where one partition corresponds to the 3Di alignment, using the $3DI + F + R6$ substitution model and the other to the amino acid alignment, using the BIC-selected most suitable substitution model for each corresponding amino acid alignment (famsa3di: $WAG + F + R9$, FoldMason: $WAG + F + R8$). Branch support was assessed using 10,000 replicates of the ultrafast bootstrap approximation method (Hoang et al, 2018). Phylogenetic trees were visualized and prepared for publication with the ggtree R package v3.10.1 (Yu et al, 2017) (Appendix Fig. S7). The phylogeny that had a topology most consistent with taxonomy (famsa3di alignment, $3DI + AA$) is presented in Fig. 3C. All accompanying structure models were visualized and prepared for publication with UCSF ChimeraX (Meng et al, 2023).

## Phylogenetic analysis of the DNA polymerase from *Herpesvirales* and *Baculoviridae*

In order to survey class-I and III fusion proteins across the *Herpesvirales* and *Baculoviridae*, we reconstructed polymerase-based phylogenies that recapitulate continuous underlying evolutionary histories, onto which we can map glycoprotein distribution. We retrieved DNA polymerases from each species in the two families based on our hallmark gene Foldseek search, using the Phi29 DNA polymerase B (PDB ID: 2PY5 (Berman et al, 2007)) as a

query. One match with the highest similarity was used for each species, resulting in 119 *Herpesvirales* and 94 *Baculoviridae* PolB amino acid sequences. Each set of sequences was aligned using mafft v7.525 (Katoh and Standley, 2013) (--localpair option) and phylogenies were inferred using IQ-TREE v.2.3.6 (Minh et al, 2020) under the most suitable substitution model selected by ModelFinder (Kalyaanamoorthy et al, 2017).

## Foldseek homology searches against class-I and III fusion proteins in the Herpesvirales and Baculoviridae

We performed Foldseek searches against the entire Viro3D database using experimentally determined protein structures for class-I and III fusion glycoproteins (RSV F pre-fusion PDB ID:5EA3 (Battles et al, 2016), RSV F post-fusion PDB ID:6APB (Goodwin et al, 2018), hMPV F PDB ID:7TJQ (Rush et al, 2022), SARS-CoV-2 S PDB ID:6VXX (Walls et al, 2020), baculovirus gp64 PDB:8YG6 (Guo et al, 2024), VSV G PDB ID:5I2F (Roche et al, 2007), HHV-5 gB PDB ID:7KDP (Liu et al, 2021)). We examined the results of this for the presence of proteins from each of the species represented in our *Herpesvirales* and *Baculoviridae* phylogenies, extracting Foldseek *e*-values or assigning an arbitrarily large *e*-value (10) when no protein was detected. *E*-values were plotted against corresponding phylogenies using iTOL (Letunic and Bork, 2024).

## Development of the Viro3D web-resource

The frontend of Viro3D was built using React and Typescript, providing a dynamic user interface. To enhance specific features, we integrated PDBE-Molstar Viewer to provide an interactive 3D rendering of protein structures (https://github.com/molstar/pdbe-molstar); Sodaviz (https://github.com/sodaviz) to allow for browsing protein structures across the virus genome; KonvaJS to construct the interactive map of viruses based on structure similarities (https://github.com/konvajs/konva). The Viro3D backend was developed using FastAPI to allow for programmatic access to the data. Biopython was integrated for leveraging the NCBI BLASTp command-line wrapper to enable searching by protein sequence. The MongoDB Community Edition was used to store the data. The source code for the web resource can be found at https://github.com/centre-for-virus-research/viro3d-frontend and at https://github.com/centre-for-virus-research/viro3d-backend.

## Data availability

The datasets and computer code produced in this study are available in the following databases: Protein structure predictions: Viro3D database (https://viro3d.cvr.gla.ac.uk/). Structural analysis code: GitHub (https://github.com/ulad-litvin/viro3d-analysis). Web resource code (frontend): GitHub (https://github.com/centre-for-virus-research/viro3d-frontend). Web resource code (backend): GitHub (https://github.com/centre-for-virus-research/viro3d-backend). All ColabFold and ESMFold structure predictions, associated metadata, Foldseek database with structural clusters, and network metadata: Zenodo (https://doi.org/10.5281/zenodo.15622906 and https://doi.org/10.5281/zenodo.15745595).

The source data of this paper are collected in the following database record: biostudies:S-SCDT-10_1038-S44320-025-00147-9.

## Peer review information

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

## Acknowledgements

This work was funded by UK Medical Research Council (MRC) support to the MRC-University of Glasgow Centre for Virus Research (CVR Integrative Viral Genomics and Bioinformatics Platform: MC_UU_00034/5, CVR Preparedness Platform: MC_UU_00034/6, and CVR Structure-to-Function of Virions Programme: MC_UU_00034/1). This research is jointly funded by the MRC and the Foreign Commonwealth and Development Office (FCDO) under the MRC/FCDO Concordat agreement. J.G. was supported by a Wellcome Trust and Royal Society Sir Henry Dale Fellowship (107653/Z/15/Z) and an Emerging Leaders Prize from the Medical Research Foundation (MRF-ELP-VAH-23-107). U.L. is supported by a PhD studentship from the Darwin Trust of

Edinburgh. We thank Prof. Emma Thomson and Prof. Massimo Palmarini for their support of Viro3D.

## Author contributions

**Ulad Litvin**: Conceptualization; Data curation; Software; Investigation; Visualization; Methodology; Writing—original draft; Writing—review and editing. **Spyros Lytras**: Conceptualization; Data curation; Investigation; Methodology; Writing—review and editing. **Alexander Jack**: Data curation; Software. **David L Robertson**: Supervision; Funding acquisition. **Joseph Hughes**: Conceptualization; Data curation; Software; Supervision; Methodology; Writing —review and editing. **Joe Grove**: Conceptualization; Supervision; Funding acquisition; Investigation; Visualization; Methodology; Writing—original draft; Writing—review and editing.

Source data underlying figure panels in this paper may have individual authorship assigned. Where available, figure panel/source data authorship is listed in the following database record: biostudies:S-SCDT-10_1038-S44320-025-00147-9.

## Disclosure and competing interests statement

The authors declare no competing interests.

# Expanded View Figures

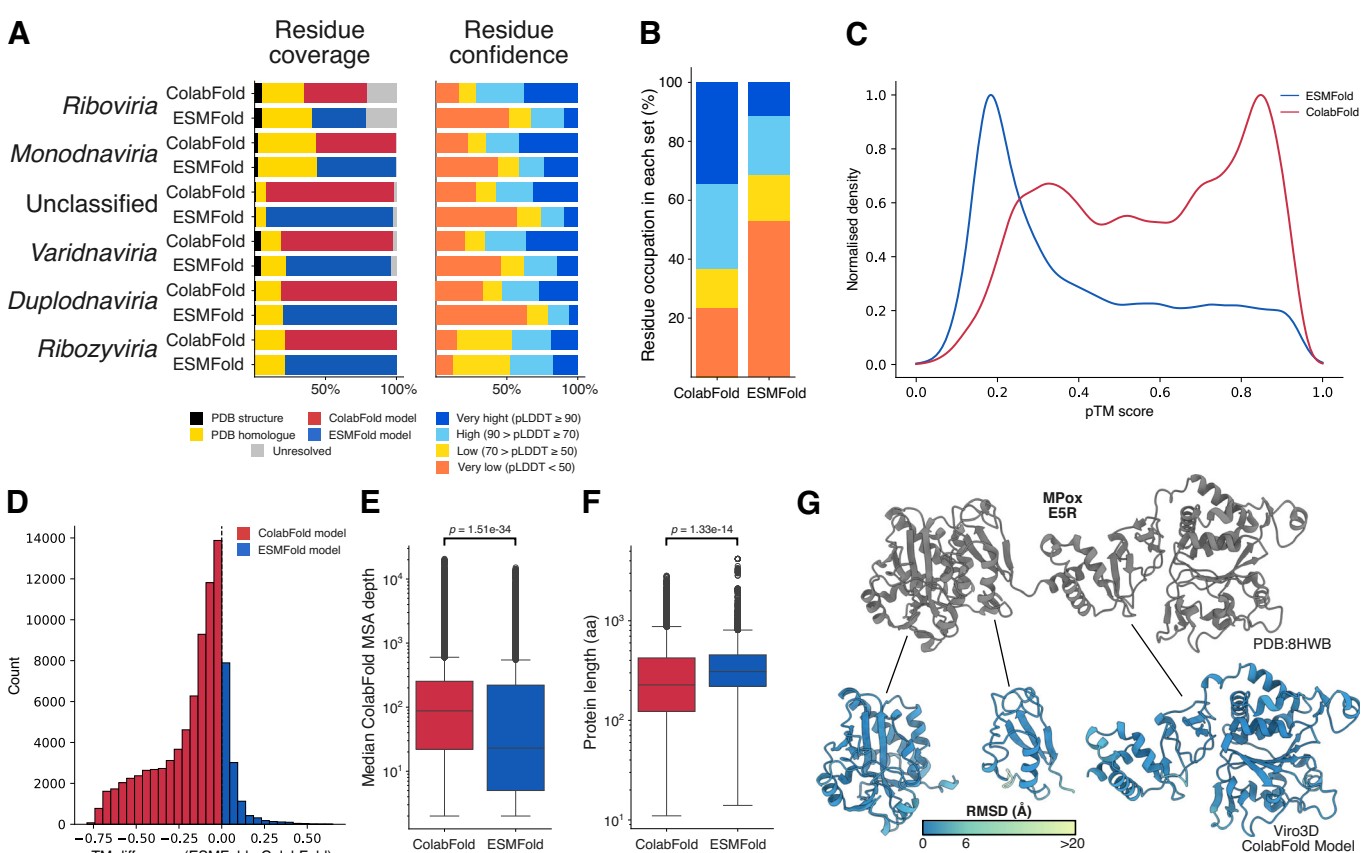

**Figure EV1.  Expanding structural coverage of the human and animal virosphere.**

(**A**) On the left, percentage of residues covered by ColabFold (red) and ESMFold (blue) models in contrast to percentage of residues covered by PDB structures (sequence identity ≥ 95%, black) and PDB homologs (sequence identity ≥30%, yellow) for each viral realm. On the right, confidence of residues modeled by ColabFold and ESMFold based on pLDDT score per viral realm. (**B**) Percentage of ColabFold and ESMFold residues with very high (dark blue), high (light blue), low (yellow) and very low (orange) pLDDT score. (**C**) Normalized distribution of pTM scores for ColabFold (red, median pTM score of 0.59) and ESMFold (blue, median pTM score of 0.31) models. (**D**) Distribution of differences in pTM scores between ColabFold and ESMFold models for each protein record where both models are available. Positive values indicate that ESMFold model is better (blue bars), negative value – ColabFold model is better (red bars). 15.74% of protein records have a higher pTM score with ESMFold prediction, 848 of these records have difference in pTM score greater than 0.2. (**E**) Distributions of median ColabFold MSA depth for records where ColabFold model has a higher pLDDT score (red box) and ESMFold models has a higher pLDDT score (blue box). The difference between MSA depth is significant based on the *t* test results (t-statistic: -12.26; *p*-value: 1.51e-34). (**F**) Distributions of protein length for records where ColabFold model has a higher pLDDT score (red box) and ESMFold models has a higher pLDDT score (blue box). The difference between protein length is significant based on the *t* test results (t-statistic: -7.70; *p*-value: 1.33e-14). In both (**E**) and (**F**), the bounds of boxes represent the interquartile range (IQR, 25th–75th percentile), with the center line showing the median (50th percentile). Whiskers extend to the minima and maxima within 1.5 × IQR. Individual data points outside the whiskers are outliers; ColabFold ($n = 77,071$), ESMFold ($n = 7769$). (**G**) Mpox E5R protein, as shown in Fig. 1G, but with the three constituent individual domains color-coded by RMSD (Å) after their respective rigid-body alignment with the experimental structure (gray). This indicates that whilst their relative positions may diverge from the experimental structure (Fig. 1G), the individual domains are accurately predicted.

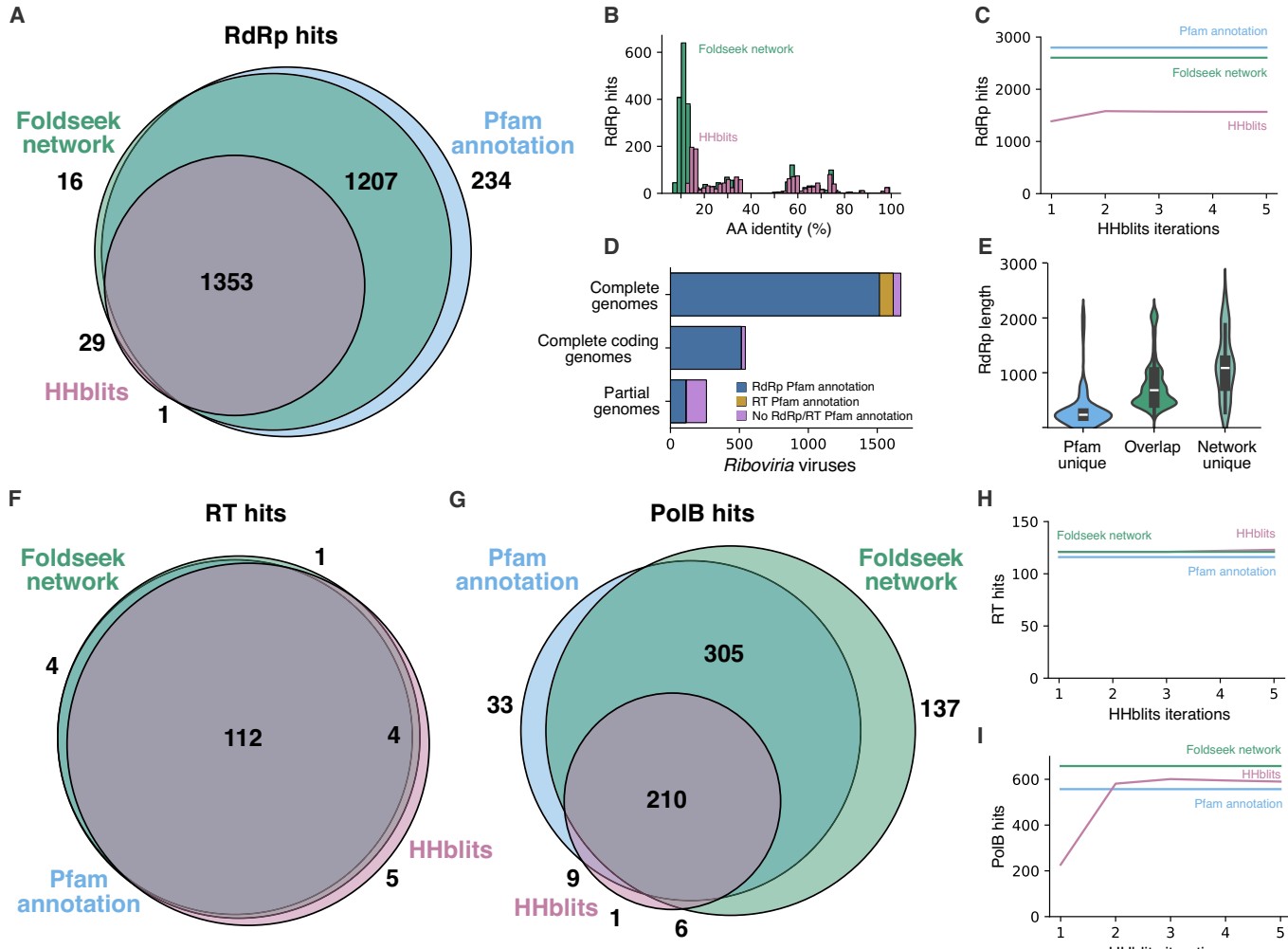

**Figure EV2. Comparison of structure and sequence-based mapping of viral polymerases.**

(**A**) Venn diagram demonstrating the overlap between the number of proteins with RdRp Pfam annotation (in light blue color) and RdRp hits found using one iteration of HHblits search (in pink) or Foldseek search against structural network (in green). (**B**) Distribution of pairwise amino acid identities between RdRp probe (PDB ID: 4ROE) and Viro3D entries identified through one iteration using HHblits (pink) or Foldseek network search (green). (**C**) Number of RdRp hits found after five HHblits search iterations in comparison to the number of records with RdRp Pfam annotation and records found using Foldseek search against structural network. (**D**) Number of viruses in the *Riboviria* that possess proteins with RdRp or RT Pfam annotation split into groups based on sequence entry genome completeness. (**E**) Distribution of protein length for RdRp records unique to Pfam annotation, Foldseek network search or present in both. The white dot in the center of the violin represents the median of the data (50th percentile). The thick bar shows the interquartile range (IQR, 25th–75th percentile). The thin line extending from the box shows the range of the data within 1.5 × IQR. The tips of the violin represent the minimum and maximum values in the data, as far as the density estimate extends. Pfam unique ($n = 238$), overlap ($n = 2560$), network unique ($n = 45$). (**F**) Venn diagram highlighting the overlap between the number of proteins with RT Pfam annotation and RT hits found using one iteration of HHblits search or Foldseek search against the structural network. (**G**) Venn diagram showing the overlap between the number of proteins with PolB Pfam annotation and PolB hits found using one iteration of HHblits search or Foldseek search against structural network. (**H**) Number of RT hits found after five HHblits search iterations in comparison to the number of records with RT Pfam annotation and records found using Foldseek search against structural network. (**I**) Number of PolB hits found after five HHblits search iterations in comparison to the number of records with PolB Pfam annotation and records found using Foldseek search against structural network.

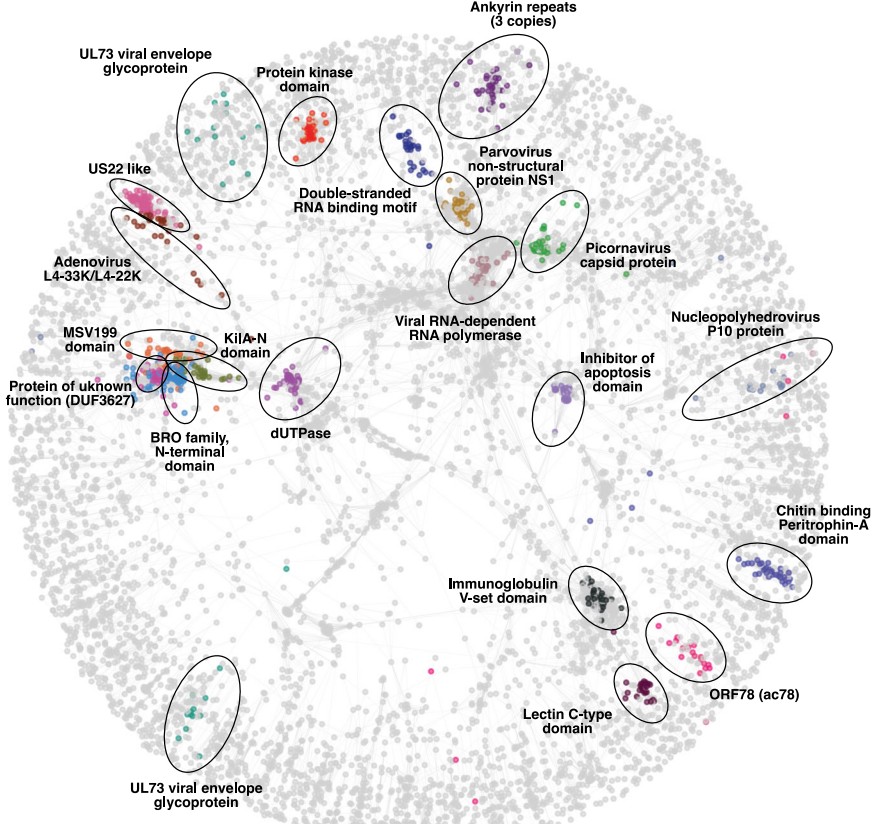

**Figure EV3.  Functionally annotated structure similarity network of viral proteins.**

Each node represents a cluster. Edges connect clusters that share structural similarity. Clusters that possess the twenty most frequent Pfam annotations are highlighted in different colors.

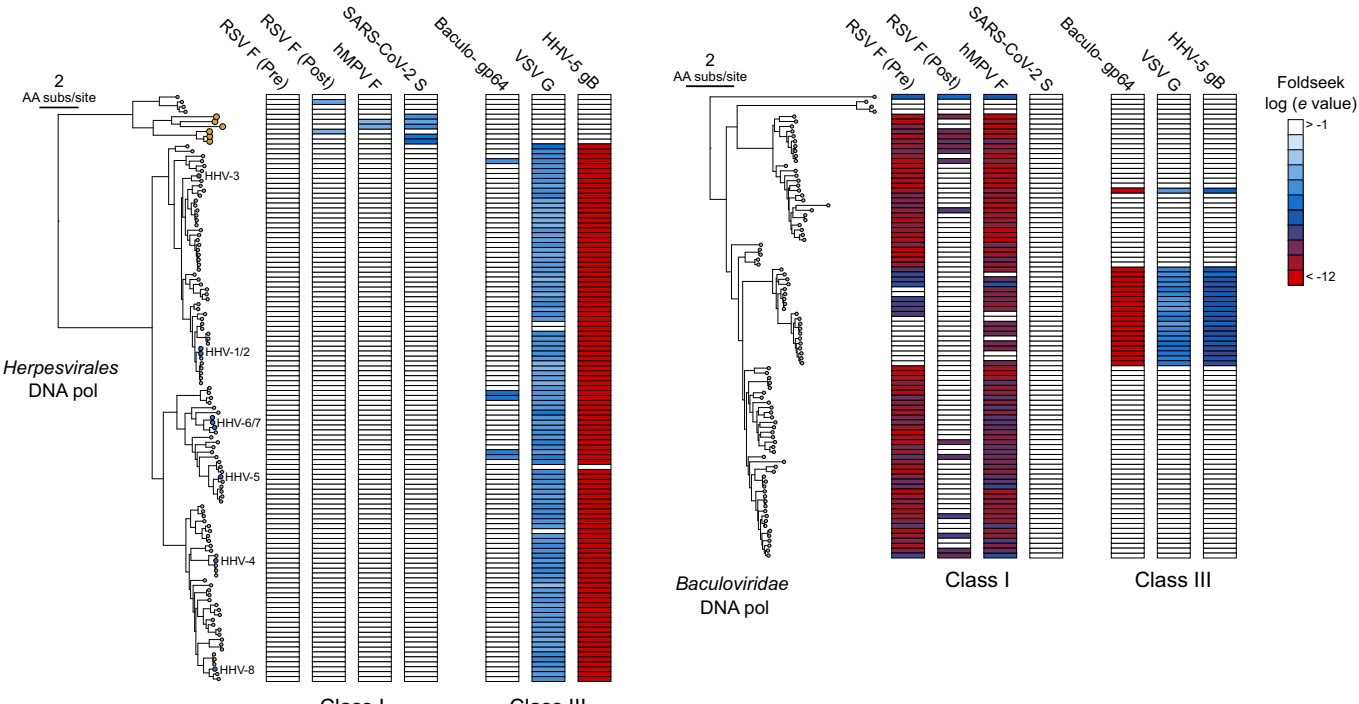

**Figure EV4. Foldseek homology for class-I and class-III fusion proteins in the *Herpesvirales* and *Baculoviridae*.**

Whole proteomes for all species within either taxonomic group were surveyed, using Foldseek, for structural homology against class-I and class-III fusion proteins. Foldseek structural homology scores were mapped against the underlying DNA polymerase amino acid sequence phylogeny to reveal the distribution of fusion mechanisms. Presented phylogenies are midpoint rooted. Heatmaps display Foldseek log transformed e-values (as indicated in the key) for the stated references, including RSV F protein in both pre- and post-fusion states (see Methods for details). For orientation in the *Herpesvirales* phylogeny, human herpesviruses have tips colored blue with species labels, whereas the aquatic herpesviruses identified in our structural cluster search (Fig. 3B) are colored yellow. Scale bar represents amino acid substitutions per site. hMPV human metapneumovirus, VSV vesicular stomatitis virus, HHV-5 human herpesvirus-5.

