## [Peer Review File · Molecular Systems Biology]

Viro3D: a comprehensive database of virus protein structure predictions

Ulad Litvin, Spyros Lytras, Alexander Jack, David Robertson, Joseph Hughes, and Joe Grove

Corresponding author(s): Joe Grove (joe.grove@glasgow.ac.uk)

Review Timeline:

Submission Date:	11th Apr 25
Editorial Decision:	17th Apr 25
Revision Received:	3rd Jul 25
Editorial Decision:	11th Aug 25
Revision Received:	26th Aug 25
Accepted:	1st Sep 25

Editor: Poonam Bheda

Transaction Report:

17th Apr 2025

Manuscript Number: MSB-2025-13039-T

Title: Viro3D: a comprehensive database of virus protein structure predictions

Dear Dr. Grove,

Thank you for transferring your manuscript to Molecular Systems Biology. As previously discussed, we will send a revised version of the manuscript back to the original reviewers, noting that for publication in Molecular Systems Biology we would not require follow-up experimental validations or mechanistic insight as requested by Reviewer 2.

When submitting your revised manuscript, please supply the following documents and information:

- 1) A .docx formatted version of the manuscript text (including legends for main figures, EV figures and tables). Please make sure that the changes are highlighted to be clearly visible. Alternatively you may choose to submit your manuscript as a LaTeX file.
- 2) Individual production quality figure files as .eps, .tif, .jpg (one file per figure). For guidance, download the 'Figure Guide PDF' (<https://www.embopress.org/page/journal/17574684/authorguide#figureformat>).
- 3) At EMBO Press we ask authors to provide source data for the main figures. Our source data coordinator will contact you to discuss which figure panels we would need source data for and will also provide you with helpful tips on how to upload and organize the files.
- 4) A .docx formatted letter INCLUDING the reviewers' reports and your detailed point-by-point responses to their comments. As part of the EMBO Press transparent editorial process, the point-by-point response is part of the Peer Review File (PRF), which will be published alongside your paper.
- 5) A complete author checklist, which you can download from our author guidelines (<https://www.embopress.org/page/journal/17574684/authorguide#submissionofrevisions>). Please insert information in the checklist that is also reflected in the manuscript. The completed author checklist will also be part of the PRF.
- 6) Please note that all corresponding authors are required to supply an ORCID ID for their name upon submission of a revised manuscript.
- 7) It is mandatory to include a 'Data Availability' section after the Materials and Methods. Before submitting your revision, primary datasets produced in this study need to be deposited in an appropriate public database, and the accession numbers and database listed under 'Data Availability'. Please remember to provide a reviewer password if the datasets are not yet public (see <https://www.embopress.org/page/journal/17574684/authorguide#dataavailability>).

In case you have no data that requires deposition in a public database, please state so in this section as follows: "This study includes no data deposited in external repositories". Note that the Data Availability Section is restricted to new primary data that are part of this study.

If you have any questions, please don't hesitate to contact me.

Yours sincerely,

Poonam Bheda, PhD
Scientific Editor
Molecular Systems Biology

Point-by-point response to reviewers comments for “Viro3D: a comprehensive database of virus protein structure predictions”

We thank the reviewers for their constructive critique of our work. We provide a point-by-point response addressing and/or discussing each of the reviewers' concerns. Reviewer remarks are shown verbatim, in grey text, our responses are provided in black text.

Many minor changes were made throughout the manuscript, we do not provide an exhaustive list of these, but do refer to substantive changes by reference to line numbers and/or figures, and highlight these in the resubmitted manuscript text. We have performed many new analyses which are provided in revised Figures. 2, 3, EV1, S7 and new Figures EV2 and EV4.

Reviewer #1 (Remarks to the Author):

The manuscript is about the modeling of vertebrate virus proteins and the creation of a public database: Viro3D, to make it accessible to the scientific community. This database is of high quality and fills an important knowledge gap (as AlphaFoldDB has excluded viral proteins). It will be essential in the growing role of structural modeling in virology.

There has been two attempts to large scale modeling of viral protein: one by Jennifer Doudna's, and another by Martin Steinegger's group.

Viro3D offers a huge improvement for two reasons:

1- Reference set: The authors use the virus metadata resource (VMR) of the International Committee on Taxonomy of Viruses (ICTV). This reference set is by far the best reflecting viral diversity, as it was created by a large community of experts and contains references representing species and sequence diversity. This is a significant improvement over previous datasets that used Refseq (about one sequence per species) or Uniref (many fragments) as a reference.

2- Methodology: The authors compared Colabfold and ESMfold and found optimal parameters that lead to optimal quality of the modeled structures.

In addition to building the database, the authors also carried out a cluster and network analysis of the viral structures. This original and promising approach enables the propagation of annotations and sheds some light on the viral dark proteome. The authors also investigate the structural evolutionary history of class I fusion proteins with an original and pioneering method.

1.1 We are grateful that the reviewer sees the value in our work and thank them for their complements.

The paper is well written and pleasant to read.

I suggest that it be accepted for publication with minor revisions.

Suggested improvements:

line 140: The text does not define a sequence homolog. It might be clearer to mention here that homolog means at least 30% similarity, as stated in Mat and Method.

We have corrected this, defining homologs wherever appropriate in the main text and figure legends.

line 204: The choice of $1e-5$ as the threshold for the e-value is extremely important for the rest of the analysis and it would be nice to discuss this. Why was this number chosen? Because of the usual blast threshold or otherwise? Would the conclusions be different with a different E value?

1.2 The threshold of $1e-5$ was a pragmatic choice that attempts to balance sensitivity with accuracy (i.e. minimising detection of “off-target” hits); we provide a description of this in the Methods (lines 591-594). We also provide specific discussion around sensitivity and accuracy in the context of comparing Foldseek network analyses and Pfam annotations, and how this could be explored using alternative clustering parameters (lines 281-286).

paragraph lines 239-252: The analysis would benefit from additional context. Realm classification by ICTV have been defined by the presence of Virus Hallmark Gene (VHG).

- RNA-directed RNA polymerases (RdRps);
- RNA-directed DNA polymerases/reverse transcriptases (RTs) that are homologous to RdRps;
- single jelly-roll major capsid proteins (SJR-MCPs);
- double jelly-roll major capsid proteins (DJR-MCPs); and
- rolling-circle replication initiation endonucleases (RCREs).

...

Riboviria is defined by ICTV as RdRp/RT-containing viruses <-> the authors identify RdRp in 91% of them using a single poliovirus reference. That's pretty good, how could it be improved?

In the same paragraph, the analysis of polB in different virus families should be compared to what is already known. A simple way to compare is to look at the Pfam PolB family

<https://www.ebi.ac.uk/interpro/entry/pfam/PF00136/taxonomy/uniprot/#sunburst>, which includes Baculoviridae and Bidnaviridae. (There is a typo in the line: Bi(n)dnaviridae).

dsDNA viruses are mainly classified by ICTV on the basis of their capsid folding. This confirms the results and it would be better to add this context. Double jelly roll fold defines Varidnavira, HK97 fold defines duplodnaviria.

1.3 We provide extensive additional discussion and contextualisation on the importance of hallmark proteins for virus classification and how this relates to our findings (231-286). We also perform new analyses cross comparing Pfam, HHblits

and Foldseek methods for the detection of hallmark proteins (new Fig. EV2). In response to Reviewer 4 (point 4.2) we describe our analyses on the success rate of RdRp detection, and include this in the revised text (lines 259-267).

Lines 262-263: Gene3d and the superfamily reference are given in mat/methods, but not in the text.

1.4 This has been corrected, thank you.

line 288- The analysis is interesting, it should be noted that most Monodnaviria and all Robozviria have circular genomes and therefore no beginning or end of the viral genome. The same is true for Adenoviridae and Adintoviridae, which also have circular genomes. Perhaps you should change the sentence to "start or end of linear viral genomes".

1.5 This is an important clarification, we have added the suggested text.

line 341: Metaviridae are retrotransposons. This is important for the analysis and should be stated. The authors have chosen to form a small group with them, which may be justified by the ancestral nature of retrotransposons.

1.6 Thank you for pointing this out. We have specified the nature of the *Metaviridae* in the text. Their grouping within Fig. 3B & C, however, are driven by data analysis and reflect their structural-level similarity between themselves and other viruses.

line 347: The Carmotetraviridae are not known to be enveloped or to encode a fusion protein (<https://ictv.global/report/chapter/carmotetraviridae/carmotetraviridae>). The authors should explain how they have included this virus family in the class I tree of fusion proteins.

1.7 In considering critique from Reviewer 2 around model accuracy, we implemented a pLDDT confidence metric filter in our phylogenetic analysis of class-I fusion proteins. Whilst this did not change any of the major conclusions, we lost 8/259 identified structures, including the protein from the *Carmotetraviridae*, which may have been erroneously identified as a homolog. We include a short note on this in the Methods section.

line 360: Very exciting observations, but be careful with this conclusion: extinct and unknown viruses are missing from the tree. Corona and allo-herpesviruses could have acquired class I from an unknown third partner.

1.8 We have revised and nuanced our interpretation of this discovery in the Discussion (lines 510-513).

line 400: Riboviria is defined by the presence VHG RdRp or RT, whose active site is homologous in the sequence and can be aligned: I would remove or correct "even where underlying sequence homology may be negligible"

1.9 Thank you for the clarification. We have removed this text.

line 402: The first sentence is incorrect. Each viral kingdom does not represent an independent origin of an evolutionarily distinct group of viruses. They are arbitrarily defined based on the presence of a VHG with some concern about DNA form (Adnaviria). In other words, one gene and some context defines each realm: the evolutionary history of viruses is much more complex. It could be more accurate to write a sentence stating that horizontal gene transfer is common in viruses.

DJR MCP and HK97 MCP are specific to realms, because that is how Varidnaviria and Duplodnaviria were defined by ICTV. If ICTV had chosen polB to define a realm, the capsids would have been distributed differently. The author's observation thus confirms the classification, and lines 402-412 may need to be revised in this sense.

1.10 In line with the revisions made to address point 1.3, we have completely revised this section of the Discussion (lines 462-479) to better reflect the importance of hallmark proteins in virus classification, how our data supports this, and the complexities introduced by horizontal gene transfer.

Thank you!

Philippe Le Mercier

Reviewer #2 (Remarks to the Author):

In this work, the authors created a large database of predicted structures of viral proteins from 4,400 animal viruses. The dataset is expected to be of broad interest and to be useful for various works in structural biology and virology. However, very few advanced analyses and no experiments were done, beyond running Alphafold and providing the structures. Running these pipelines can be done and has been done by various groups (the authors mention two similar studies already published - Nomburg et al. and Kim et al, and there are several others). The few further analyses done using the initial data are mostly technical in nature and are neither deep or comprehensive. In my opinion, they lack novel insights with regards to viral protein evolution, viral replication and/or interactions with the host. Without significant additional analyses providing such insights (e.g., estimates of acquisition times of various proteins, analysis of macro- and micro- changes in the structures of proteins between different viruses and between viral and host proteins and how these can be related to viral physiology, conservation of function and cases of neo-functionalization, etc.) or without further mechanistic experimental studies, this work seems more suitable as a resource in a database section of an omics or a bioinformatics journal.

Several structural analyses performed across the dataset, such as the estimation of structural coverage relatively to experimentally solved structures and the structural clustering using FoldSeek are important to test the data structure and quality, but are mostly technical, and lack the novel biological insights I mentioned (e.g., the increase in identification of RdRp proteins is a proof-of-concept of the potential use of this dataset and is not followed by additional analyses or experimental validations).

The analysis at the end of the manuscript, of class-I fusion glycoproteins, is rather shallow and, in my opinion, does not provide important new insights. It lacks comparison to phylogeny-based sequence evolution analysis and the conclusion regarding the origins of the coronavirus spike glycoprotein is speculative and is not supported by sufficient analyses.

2.1 We appreciate the reviewer's concerns about the extent and depth of the analyses presented in this manuscript. We expect that Viro3D (and other similar resources) will permit comprehensive investigations of the deep evolutionary history of many viral proteins. We provide initial analyses of an example system, class-I fusion proteins, to demonstrate what can be achieved with Foldseek analyses using only a single reference structure. This has provided surprising new insights (e.g. class-I fusion proteins in *Herpesvirales*) that warrant further investigation, including lab-based experiments. However, we suggest that these detailed investigations fall outside of the scope of this work, which serves primarily as a technical demonstration of what is possible. Nonetheless, we have performed further Foldseek analyses on the distribution of class-I and class-III fusion proteins in the *Herpesvirales* and *Baculoviridae* (Fig. EV4, lines 374-389 & 486-496), which provide further evidence to support our conclusions.

In addition, I have concerns regarding the ability to efficiently use this data and to reproduce the analysis, as follows:

While the online website provides the ability to search for specific viruses and specific proteins, in no place one can find the full list of viruses and proteins, and there is no place where the entire list of all viruses and all proteins can be easily downloaded from – severely hampering the ability of computational structural biologists interested in downloading all the data, or understanding what the data is composed of, to use it.

I would also suggest to add supporting tables with the lists of all viruses and proteins and their identifiers to the manuscript. This is important for both reproducibility of this work as well as to help future users in finding their virus / protein of interest.

The authors made several analyses where they claim that they expanded the functional coverage for some proteins with missing annotations – these new annotations should also appear as a table for future use by readers.

In subsequent analyses (Fig 2-3), it is not clear whether the authors used all the predicted structures (the highest ranked model from each) or whether only models with pLDDT score above 0.7, which are generally considered reliable, were used. In any case, a list of all proteins with their average scores should appear in the supporting information.

2.2 The reviewer is correct, the Viro3D web portal alone does not meet the highest standards of data availability and sustainability. To rectify this, we provide all predicted structures (ColabFold and ESMFold) and their associated data (full confidence metrics, multiple sequence alignments, Foldseek database and structural clustering information) across two Zenodo repositories (Part 1: 10.5281/zenodo.15622906 and Part 2: 10.5281/zenodo.15745595). We have also

provided various supporting information (Appendix Supplementary Tables 1-6) containing information on: 1) viral species 2) individual proteins 3) members of structural clusters 4) connections between clusters 5) lists of expanded Pfam annotations and 6) class-I fusion glycoproteins identified in our analyses.

We have provided details on model selection in the Methods section (we used only the top ranked models from ColabFold). Consistent with all other similar resources (e.g. AlphaFold Database and Big Fantastic Virus Database) we do not filter structures by confidence metrics. By sharing all models, we maximise data transparency and permit investigation of features such as protein disorder (which correlates with low confidence metrics). We have, however, included confidence filtering for the phylogenetic reconstruction of class-I fusion proteins (loosing 8/259 proteins). The revised phylogeny is provided in Fig. 3C, this did not alter any of the major conclusions of this analysis.

Additional comments:

The term “mature peptide” when referring to the final cleaved protein products originating in polyproteins should be more accurately described as “matured cleaved protein”.

In Methods, please provide additional details regarding how polyproteins were analysed beyond the current details.

2.3 We have provided clarification in the main text and further details in the Methods. It is important to note, however, that “mature peptide” is the GenBank annotation type used for these proteins (lines 121-123 & 534-535).

In the Reporting Summary, there is a strange identifier following “FoldMason” – please clarify, remove or change:

“Structure-guided multiple sequence alignments (MSAs) were generated using FoldMason v99fbda50e6c296c9fdf15d05fefedbb91f4efe84”

2.4 This represents the version number of FoldMason, we have removed this from the main text, but provide it in the Tools Table.

The article suffers from grammatical errors and typos in various sections, e.g.

In Methods:

“For records with protein length greater than 1,236 aa following settings were used...”

In Abstract:

“Tolerance to genetic change, high mutation rates, adaptations to hosts and immune escape has driven extensive...”

2.5 Thank you for highlighting these errors, we have made grammatical corrections and improvements throughout the entire text.

Reviewer #3 (Remarks to the Author):

The manuscript by Litvin et al. describes the prediction of virus-encoded proteins from the list of ICTV recognized viruses. The predicted protein structures were evaluated for quality by the average pLDDT score and structural clustering was performed to infer evolutionary relatedness. All data is available through an easy-to-use web interface that will be a great resource for the community. The manuscript is well written and easy to follow. There are, however, some important concerns that the authors need to address.

3.1 We thank the reviewer for their positive comments.

Major:

Using the average pLDDT score as a quality measure for a prediction is misleading, as there are two main reasons for a low mean pLDDT in AlphaFold. The first is an underpopulated MSA which results in low co-evolution signal and bad folding. The second however can be the fact that large parts of a protein are disordered. In fact, several protein disorder databases use pLDDT as a predictor for IDRs and the AlphaFold FAQ explicitly states that a low pLDDT is an indicator for protein disorder. Therefore, the authors need to either analyze both MSA and pLDDT or use another score such as pTM as a quality score.

3.2 We thank both Reviewer 3 and Reviewer 4 for their similar suggestions around including additional confidence metrics. We agree that pTM score is an important metric for assessing global structure quality, however, we would like to point out that it can be affected by flexible disordered linkers or uncertain domain orientations in multi-domain proteins. In contrast, average pLDDT score better reflects local confidence, particularly in individual domains. Since both metrics provide complementary insights, we plotted the distribution of ColabFold and ESMFold pTM scores (Fig. EV1C) and pTM score difference between both predictions (Fig. EV1D). ColabFold has a median pTM score of 0.59, while ESMFold has 0.31. Interestingly, 15.74% of protein records have a higher pTM score with ESMFold prediction, 848 of these records have difference in pTM score greater than 0.2. Unfortunately, since pTM is a per model score, Figure 1D cannot be repeated with pTM scores because it represents a per residue pLDDT distribution (not a distribution of average pLDDT scores). We describe and discuss these results in the Results (lines 144-165).

We also agree that investigators may wish to evaluate protein disorder using Viro3D structures and, in this context, analyses of underlying multiple sequence alignments may be beneficial. These resources are now fully available via the Zenodo repositories associated with this manuscript.

In any case, clear quality thresholds need to be established and used as there is no quantitative analysis of a high or low quality prediction. The proteins that are low quality should be predicted again with ColabFold with more recycles and the original AlphaFold from DeepMind that uses jackhammer for MSA generation. Were templates used? In addition, the number of recycles used should be stated in the methods. Still low-scoring models need a warning note on the website or should be deleted.

3.3 We provide additional details of our prediction approach in the Methods (lines 546-549): for each protein record ColabFold produced 5 models with 3 recycles and no PDB templates. ESMFold produced 1 model using 4 recycles. As discussed in response to Reviewer 2, we assert that sharing all models is good for data transparency and this is consistent with the approach of other large repositories. Moreover, it is very difficult to define appropriate confidence thresholds by which to exclude structures; these will likely vary depending on the needs of the downstream user. For example, proteins with well-folded domains and large, functionally important, disordered segments (e.g. NS5A in the hepaciviruses) will have lower confidence scores.

Moreover, the Viro3D web resource includes a Limitations and Feedback section that states: “It is important to remember that the structures are predictions and need to be interpreted with care. Each model comes with confidence metrics (pLDDT and pTM) and are shown in the molecular viewer with residues colour-coded by pLDDT score.”

We agree that sequences that yield low confidence models will benefit from further structure prediction efforts, including the use of alternative state-of-the-art approaches (e.g. Boltz-1, Chai-1). We plan to integrate these improvements on future iterations, but consider this beyond the scope of the first phase of Viro3D.

Signal peptides of transmembrane proteins are currently not marked. The user of the website might confuse a signal peptide with a transmembrane helix. The authors need to either mark them or provide model where the signal peptide is deleted.

3.4 This is a limitation of systematic at-scale structure prediction and is common to other large databases (e.g. AFDB and BFVD). We are committed to the future development of Viro3D, and domain-based annotation is an obvious enhancement. However, this will take some time and thought to implement properly, particularly when using systematic sequence-based predictions of signal peptides and transmembrane domains.

The BFVD from the Steinegger lab (doi.org/10.1093/nar/gkae1119) was generated using an optimized pipeline to generate deeper MSAs to increase prediction accuracy. Fig. S2 omits this comparison. The authors need to compare prediction accuracy of the proteins found in both Viro3D and BFVD using both pTM and local pLDDT values. Of note the pLDDT confidence difference between Nomburg et al and Viro3D is not surprising, as Nomburg et al. used an older version of AlphaFold. The same would likely happen if the Viro3D database had been run in AlphaFold3.

In general, the authors need to do a systematic comparison of Viro3D to BFVD, not just pick certain viral classes. Figure S2 D,E should compare the pTM. Figure S2 F does not have a structure for BFVD influenza NA although the BFVD contains influenza NA proteins.

3.5 First, we acknowledge that the Nomburg et al. dataset was generated using an older version of ColabFold, v.1.3.0 (we used 1.5.2), however, both versions generate MSA and protein structure in the same way, and with the same model weights. The

difference in quality of the predictions most likely comes from the fact that: (1) Nomburg et al. used the RefSeq virus protein database for MSA assembly, instead of native ColabFold environmental database, (2) generated only 3 models per protein record while we produced 5, and (3) stopped the structure prediction when it reached average pLDDT score of 70 while we did not set the upper quality threshold.

We appreciate the reviewers concerns around our comparisons with BFVD, however, there are two features of BFVD that confound attempts to perform meaningful systematic comparison with Viro3D: (1) BFVD contains structures for representative sequences from UniRef30 clusters. Consequently the proteome of any given viral taxa is not well represented, and the closest homologs for any given protein may have very low sequence similarity. (2) UniRef30 is contaminated by a high number of fragmentary viral sequences, and BFVD has sampled these extensively. Therefore, there are relatively few pairs of complete homologous proteins to directly compare confidence metrics between BFVD and Viro3D. We acknowledge that BFVD may indeed achieve higher confidence metrics (the recently released BFVD v2 has increased prediction quality), however, this does not overcome its key limitation: low sequence depth and fragmentary sequence data.

Our decision to focus on one viral taxa in Figure S2F (now Appendix Figure S1) was to illustrate the major differences between the resources. Influenza A PR8 is a good example of an extensively used model system for which high-quality predicted structures may be particularly valuable to molecular virologists. We searched for sequence homologs to each of its proteins in the Nomburg et al. dataset and BFVD. Due to the limitations discussed above, many proteins lacked complete counterparts in BFVD, including neuraminidase (NA).

Examining the case of NA more closely, Foldseek server searches of BFVD using the Viro3D NA entry give a variety of hits. The highest BFVD sequence homolog (89.3% sequence identity) is a short fragment of 82 residues covering only 14% of the length of NA. Whereas, the more complete NA counterparts (covering >95% of the length of NA) are poor sequence homologs, for example this protein has only 32% sequence identity to Influenza A PR8 NA. Therefore, whilst we acknowledge that BFVD performs excellently at surveying total structural diversity across virology, it remains difficult to find many proteins that are likely to be of interest to molecular virologists.

Nonetheless, we have nuanced our discussion of both the Nomburg et al. dataset and BFVD to highlight the strengths of each resource (lines 207-20 & 430-436).

Why were the proteins clustered by sequence before being clustered by structure? Using only a representative from each sequence cluster will lead to missed connections. This is exactly what is seen in Figure 2B. The entire non-redundant structure database should be clustered by sequence similarity. Recent work has done that such as doi.org/10.1038/s41467-024-54668-2.

3.6 We focussed our analyses on structural similarity between proteins, beyond what can be easily found using sequence data alone. Therefore, it was important to ensure that our network predominantly captures the structural relationships between proteins, rather than the sequence relationships that are already identified by tools

like BLAST and MMseqs2. To ensure that this is the case, it was important to start with groups of proteins that share substantial sequence identity. Since it is well established that proteins sharing more than 30% sequence identity almost always adopt highly similar three-dimensional structures, we used strict clustering parameters of 50% sequence identity and 90% bi-directional coverage to avoid losing structural connections and keep the structural diversity of our dataset intact. Proteins that have additional domains or share low sequence identity do not reside in the same cluster ensuring that all clusters are structurally homogeneous. This also allowed us to increase the quality of analysed predictions by representing each cluster with the structure of the highest confidence.

In the case of Fig. 2B, this demonstrates that a network constructed from representatives of MMseqs2+Foldseek clusters allows us to find a higher number of similar structures than a Foldseek search against a non-redundant dataset.

Finally, Soh et al. built a network for only 844 protein structures from 9 viruses, whereas Viro3D contains hundreds of orthologous proteins with high sequence identity and almost identical structures. Our current structural network is highly complex and captures the relationships between only 19,000 highest-quality structures selected after two rounds of clustering. If we produced the structure similarity network using all 85,000 protein structures, it would predominantly reflect relationships that can be identified using sequence-based methods rather than those driven by unique structural features.

Furthermore, clustering should be performed with other methods for comparison, such as hhblits. Is a structure similarity search different? How do the authors distinguish a novel connection from a false positive? Line 276 “viral proteins do not share detectable homology with cellular life”. The homology search was only performed with Foldseek which has relatively low sensitivity. A homology search should also be performed at the sequence level (such as with hhblits) and with a more sensitive structure similarity search (such as TMalign: doi.org/10.1038/s41586-023-06583-7) before such a claim and claims like “ultra-sensitive homology detection” can be made.

3.7 Thank you to the reviewer for this excellent suggestion for a benchmarking exercise. We performed a comparison of HHblits, Foldseek (against complete dataset) and Foldseek network using nine viral hallmark proteins as probes (Fig. 2B). To not bias the comparison, we limited all searches to one iteration: (1) HHblits search was performed against 85,162 Viro3D protein sequences using e-value of $1e-3$, (2) Foldseek search - against 85,162 Viro3D protein structures using e-value of $1e-5$. (3) Foldseek network search was performed against 19,067 cluster representatives, using e-value of $1e-5$, but also included adjacent structural clusters in the network.

Even though a restrictive sensitivity threshold of $1e-5$ was used for all Foldseek searches to reduce false positive or ambiguous hits, Foldseek structural search demonstrated higher sensitivity than HHblits returning a greater number of significant hits for all hallmark probes except for single and double jelly roll (1,952 SJR HHblits hits vs 1,932 SJR Foldseek hits; 218 DJR HHblits hits vs 204 DJR Foldseek hits). Foldseek network further increased the number of significant hits found by Foldseek,

almost doubling them for some hallmark proteins (RdRp, Class III FG). This is described and discussed in the Results (lines 231-250)

A more detailed analysis was performed for RdRp, RT and PolB hits (Fig. EV2) where we compared the overlaps between Foldseek network hits, HHblits hits and Pfam annotations. HHblits search usually requires multiple iterations to reach results comparable to a single Foldseek network search. The diversity of RdRps represents an exception that can be captured only using the Foldseek network. This is described and discussed in the main text (lines 259-286).

Upon careful consideration we decide not to perform benchmarking against TM-align, as this approach is extremely slow and computationally expensive. However, we have also weakened our wording and removed claims of “ultra-sensitive homology detection”.

Reviewer #4 (Remarks to the Author):

This manuscript generated structure predictions of the ICTV genotypes and provides them through a web interface. Through specific structure similarity searches, Class I fusion glycoproteins were unexpectedly found in herpesviruses and baculoviruses. Although this is an interesting finding, this manuscript lacks orthogonal validation and should perform additional computation analyses.

The average pLDDT of a protein is not a good measure of prediction quality. A part of a protein could be structured while the majority of the amino acids are in a disordered region. The pTM is a measure of a protein structure prediction quality. The evaluation of the quality of the database should be based on the pTM. Figure 1C and D should be repeated with the pTM value.

4.1 This comment echos the concerns of Reviewer 3 around confidence metrics. The revised manuscript (Fig. EV1) and associated metadata (Appendix Table S2), now provides full description of both pLDDT and pTM confidence scores.

Only 91% of RNA viruses have a polymerase. What about the remaining 9%? An RNA virus must have a polymerase. Why were these not found?

4.2 In addressing Reviewer 1’s concerns around the detection of hallmark proteins we evaluated this in detail and include new analyses in Fig. EV2. Our dataset contains 2,475 viruses that belong to the realm *Riboviria*, however, 261 of them have only partial genomes (Fig. EV2D). Moreover, only 2,140 viruses have a protein record with Pfam RdRp annotation and 106 with Pfam RT annotation. Using a single RdRp protein structure we found significant hits in 2,141 viruses: 2,098 of them have records with Pfam RdRp annotation (Fig. EV2A). By repeating these procedures with an RT structure we found hits in 110 viruses: 106 of them have records with Pfam RT annotation (Fig. EV2F). The remaining viruses in the realm *Riboviria* have either incomplete genomes or genomes lacking extensive annotation and do not seem to possess RdRp or RT proteins. We described these limitations in the Results (lines 259-270).

The proteins were first clustered by sequence and then a representative protein was used for a structure similarity search. This analysis should be repeated where a structure similarity search is performed on all protein structures. Using a subset of proteins based on sequence clustering undermines the value of a structure prediction database.

4.3 Reviewer 3 posed the same concern around sequence clustering. In our response (3.6) we provide a detailed account of our motivations. In summary, we tailored our analyses to maximise the signal from structural similarity, while minimising the contribution of sequence similarity (which can readily be detected using standard sequence-based approaches).

Class I fusion glycoproteins were unexpectedly found in Herpesvirales and Baculoviridae. Herpesviruses typically encode a Class III fusion glycoprotein called gB, and in baculoviruses this is gp64. Is a gB and gp64 homolog missing in these viruses, respectively? What fusion glycoproteins do they contain? This manuscript should provide orthogonal validation of the identified glycoproteins, for example by demonstrating the expression and activity of these glycoproteins. Furthermore, which viruses contain the identified Class I glycoprotein as well as the identified Class I fusion glycoproteins need to be listed in a supplemental table.

4.4 We agree with the reviewer that this was a surprising and unexpected finding. Whilst lab-based experimental studies are beyond the scope of this work, we performed additional Foldseek analysis to map both class-I and class-III fusion proteins across the *Herpesvirales* and *Baculoviridae*, which we include in the new Fig. EV4. This demonstrates that class-III fusion predominates in the Herpesvirales, with only a minor divergent clade possessing a class-I fusion protein. In contrast, class-I fusion predominates in the *Baculoviridae* and gp64 (class-III) is mainly confined to group I alphabaculoviruses. We include description of this additional work in the main text (Fig. EV4, lines 374-389 & 486-496).

A structure similarity search should also be performed for Class II and Class III fusion glycoproteins. Where did the mononegavirales with a Class III fusion glycoprotein acquired it from?

4.5 We expect to use Viro3D to provide detailed investigations of the distribution and evolutionary history of important viral glycoproteins. A thorough examination will likely require multiple iterative searches against different reference structures, further phylogenetics and extensive biological interpretation. We agree that this would be a valuable effort, however, in our opinion lies beyond the focus of this initial work, which aims to establish the resource and demonstrate new capabilities.

A “download all” function should be available from a repository, such as Zenodo.

A number of features should be added to the viro3D website:

-The virus cluster map is not searchable. The interface needs to have a search function to find your virus of interest and see how it is connected to other viruses.

-The protein clusters have to be available on the website, and this must be searchable. What are the Foldseek scores?

-Add the MSA, pLDDT plot, and PAE plot on the prediction webpages.

-The signal peptide and transmembrane domains of the proteins should be annotated in the predictions and genome schematics.

-The predictions should have an evaluation of the prediction quality and state when a prediction is low quality.

4.6 We agree that the underlying data, annotations and interface of Viro3D could be improved, and we are committed to integrating various additional features including structure search and protein network navigation. However, effective user-friendly implementation of these features presents challenges and is time-consuming, therefore, we expect to achieve this in the coming year.

In the meantime, we have improved data availability and sustainability (as described in point 2.2) by providing all structures, metadata and cluster information across two new Zenodo repositories (Part 1: [10.5281/zenodo.15622906](https://zenodo.org/record/15622906) and Part 2: [10.5281/zenodo.15745595](https://zenodo.org/record/15745595)).

For many mononegavirales (e.g. rabies, measles, ebola, marburg), the L protein is cut into pieces in the genome schematic. The L protein is a single protein. The genome illustration is wrong. Many bunyaviruses (e.g. hantaan, rift valley fever) are also not illustrated accurately with redundant annotations or cleavages. Which blocks are Pfam annotations and not proteins should be made clear.

4.7 This is a consequence of pragmatic choices around our structure prediction approach. Structure inference for long proteins is computationally expensive and time consuming, for 2,087 proteins longer than 2,000 amino acids (many of which are polyproteins without mature peptide annotation) we generated models using protein region annotation. One of the undesirable but inevitable consequences of this approach was the fact that some long records like *Mononegavirales* L proteins were split into multiple fragments. We are actively seeking mechanisms to systematically improve input sequence annotations, however, this is a complex and time-consuming problem to resolve. In the Viro3D genome browser protein records are shown in grey, mature peptides in purple and protein regions (Pfam, CDD domains) in cyan.

The code uploaded to the github link provided contains information and documentation. The inclusion of the graph outputs in the jupyter notebooks made the code clear. I am satisfied with the deposited code.

Minor

The ColabFold methods does not specify whether templates were used for the predictions. This is important for when they make claims in Figure 1C about the differences between proteins with and without a homolog in the PDB.

4.8 We did not use templates for protein structure prediction. We added this information to the corresponding Methods section of the manuscript (546-549).

11th Aug 2025

Manuscript Number: MSB-2025-13039R

Title: Viro3D: a comprehensive database of virus protein structure predictions

Dear Dr. Grove,

Thank you for the submission of your revised manuscript to Molecular Systems Biology. I am pleased to inform you that we will be able to accept your manuscript pending the following final amendments and appropriate response to reviewers:

1) In the main manuscript file, please include keywords to max. 5.

2) Please format the Data availability section according to the example below:

"The datasets and computer code produced in this study are available in the following databases:

- Chip-Seq data: Gene Expression Omnibus GSE46748 (<https://www.ncbi.nlm.nih.gov/geo/query/acc.cgi?acc=GSE46748>)

- Modeling computer scripts: GitHub (<https://github.com/SysBioChalmers/GECKO/releases/tag/v1.0>)

- [data type]: [full name of the resource] [accession number/identifier] ([doi or URL or identifiers.org/DATABASE:ACCESSION])"

3) Please rename "Conflict of Interest" to "Disclosure and competing interests statement". We updated our journal's competing interests policy in January 2022 and request authors to consider both actual and perceived competing interests. Please review the policy <https://www.embopress.org/competing-interests> and update your competing interests if necessary.

4) Author contributions: Please remove it from the manuscript and specify author contributions in our submission system.

CRediT has replaced the traditional author contributions section because it offers a systematic machine-readable author contributions format that allows for more effective research assessment. You are encouraged to use the free text boxes beneath each contributing author's name to add specific details on the author's contribution. More information is available in our guide to authors:

<https://www.embopress.org/page/journal/17574684/authorguide#authorshipguidelines>

5) Our journal encourages inclusion of *data citations in the reference list* to directly cite datasets that were re-used and obtained from public databases. Data citations in the article text are distinct from normal bibliographical citations and should directly link to the database records from which the data can be accessed. In the main text, data citations are formatted as follows: "Data ref: Smith et al, 2001" or "Data ref: NCBI Sequence Read Archive PRJNA342805, 2017". In the Reference list, data citations must be labeled with "[DATASET]". A data reference must provide the database name, accession number/identifiers and a resolvable link to the landing page from which the data can be accessed at the end of the reference. Further instructions are available at .

6) The Methods and Protocols section should be renamed to "Methods".

7) Please place individual sections of the manuscript in the following order: Title page - Abstract & Keywords - Introduction - Results - Discussion - Methods - Data Availability - Acknowledgements - Disclosure and Competing Interests Statement - References - Figure Legends - Expanded View Figure Legends.

8) For the figures and figure legends, please take care of the following:

- Please make sure that figure callouts occur sequentially. Currently Fig. 3 is called out before Fig. 2D.

- Please note that the box plots need to be defined in terms of minima, maxima, centre, bounds of box and whiskers, and percentile in the legends of figures EV1 F; EV2 E

- Please note that information related to n is missing in the legends of figures 2F, EV1E, F; EV2 E

9) Appendix Tables S1-S6 are actually datasets that should be uploaded as one .xsl file per table and renamed to Dataset EV1-X. Each dataset will need its legend removed from the Appendix and added to the corresponding file in a separate tab. Please also be sure to update the source file names, the titles in the submission system and in the files themselves (in the legend), as well as their callouts in the main manuscript text. These tables should then also be removed from the Table of Contents in the Appendix file.

10) Synopsis:

- Synopsis image: Please provide a graphic that summarises the main findings of the manuscript on a glance and upload it as a high-resolution jpeg file 550 pixels wide x (300-600) pixels high.

- Synopsis text: Please provide a short standfirst (maximum of 300 characters, including space), limit the bullet points to max. 5 and upload it as a separate .doc file. Please write the bullet points to summarise the key NEW findings. They should be designed to be complementary to the abstract - i.e. not repeat the same text. We encourage inclusion of key acronyms and quantitative information (maximum of 30 words / bullet point). Please use the passive voice.

11) As part of the EMBO Publications transparent editorial process initiative (see our policy here:

https://www.embopress.org/transparent-process#Review_Process), Molecular Systems Biology will publish online a Peer Review File (PRF) to accompany accepted manuscripts. This file will be published in conjunction with your paper and will include the anonymous referee reports, your point-by-point response and all pertinent correspondence relating to the manuscript. Let us know whether you agree with the publication of the PRF and as here, if you want to remove or not any figures from it prior to publication. Please note that the Authors checklist will be published at the end of the PRF.

12) After your paper is published, we may promote it on social media. If you have any handles or hashtags for Bluesky you would like included, please let us know.

13) Please provide a point-by-point letter INCLUDING my comments as well as the reviewer's reports and your detailed responses (as Word file).

I look forward to reading a new revised version of your manuscript as soon as possible.

Yours sincerely,

Poonam Bheda, PhD
Scientific Editor
Molecular Systems Biology

Reviewer #2:

The authors have adequately addressed my questions and concerns. I have no further questions or suggestions for modifications.

Reviewer #3:

I sincerely thank the authors for their major revision of the manuscript and have no further comments. I recommend publication.

Reviewer #4:

The authors have addressed many of my concerns and added justifications and explanations to the text. My remaining criticisms are with the inaccurate annotations of certain orders on the website. It is misleading and people that use this website as a resource may not realize that the illustration is not displaying the gene organization.

In the Ebola virus Genome, the L protein is displayed as 3 pieces that look like 3 separate ORFs with 1 in an overlapping reading frame.

This would be a reasonable assumption since it looks the same as how the overlapping reading frames in the influenza A virus Genome are accurately depicted.

The color coding is very helpful, but the complete gene needs to always be displayed. I would be satisfied if every time a protein was split into multiple fragments (displayed in cyan), that the complete gene is shown above in grey, similar to what is done for mature proteins like Hantaan virus G1 and G2.

Minor

386: "Gb" should be "gB"

"influenza A virus Genome" is not capitalized but "Ebola" and "Hantaan" are. The capitalization should be consistent.

In Hantaan virus, G1 and G2 are labelled with "see comment". However there is no "comment" on the page.

Point-by-point response to editorial requests and reviewers comments for "Viro3D: a comprehensive database of virus protein structure predictions"

We thank the editor and reviewers for their input and constructive critique of our work. We provide a point-by-point response to all comments. Remarks are shown verbatim, in grey text, our responses are provided in black text.

Editorial Requests:

1) In the main manuscript file, please include keywords to max. 5.

Keywords have been provided, underneath the abstract (lines 31-32).

2) Please format the Data availability section according to the example below: "The datasets and computer code produced in this study are available in the following databases:

- Chip-Seq data: Gene Expression Omnibus GSE46748

(<https://www.ncbi.nlm.nih.gov/geo/query/acc.cgi?acc=GSE46748>)

- Modeling computer scripts: GitHub

(<https://github.com/SysBioChalmers/GECKO/releases/tag/v1.0>)

- [data type]: [full name of the resource] [accession number/identifier] ([doi or URL or identifiers.org/DATABASE:ACCESSION)]"

The Data Availability section has been updated (lines 773-785).

3) Please rename "Conflict of Interest" to "Disclosure and competing interests statement". We updated our journal's competing interests policy in January 2022 and request authors to consider both actual and perceived competing interests. Please review the policy <https://www.embopress.org/competing-interests> and update your competing interests if necessary.

This has been corrected (line 799).

4) Author contributions: Please remove it from the manuscript and specify author contributions in our submission system. CRediT has replaced the traditional author contributions section because it offers a systematic machine-readable author contributions format that allows for more effective research assessment. You are encouraged to use the free text boxes beneath each contributing author's name to add specific details on the author's contribution. More information is available in our guide to authors:

<https://www.embopress.org/page/journal/17574684/authorguide#authorshipguidelines>

This section has been removed and full author contributions provided in the submission system.

5) Our journal encourages inclusion of *data citations in the reference list* to directly cite datasets that were re-used and obtained from public databases. Data citations in the article text are distinct from normal bibliographical citations and should directly link to the database records from which the data can be accessed. In the main text,

data citations are formatted as follows: "Data ref: Smith et al, 2001" or "Data ref: NCBI Sequence Read Archive PRJNA342805, 2017". In the Reference list, data citations must be labeled with "[DATASET]". A data reference must provide the database name, accession number/identifiers and a resolvable link to the landing page from which the data can be accessed at the end of the reference. Further instructions are available at <https://www.embopress.org/page/journal/17574684/authorguide#referencesformat>

We have added reference to the ICTV Virus Metadata Resource, which was used as the basis for our sequence selection (lines 116-117, 533 ,889-890).

6) The Methods and Protocols section should be renamed to "Methods".

This has been corrected.

7) Please place individual sections of the manuscript in the following order: Title page - Abstract & Keywords - Introduction - Results - Discussion - Methods - Data Availability - Acknowledgements - Disclosure and Competing Interests Statement - References - Figure Legends - Expanded View Figure Legends.

This has been corrected.

8) For the figures and figure legends, please take care of the following:
- Please make sure that figure callouts occur sequentially. Currently Fig. 3 is called out before Fig. 2D.

We have checked and corrected the figure callouts.

- Please note that the box plots need to be defined in terms of minima, maxima, centre, bounds of box and whiskers, and percentile in the legends of figures EV1 F; EV2 E
- Please note that information related to n is missing in the legends of figures 2F, EV1E, F; EV2 E

We have added details on statistical representations and n values to all relevant figure legends:

Figure 2F: lines 1099-1106
Figure EV1 E & F: lines 1145-1155
Figure EV2 E: lines 1171-1177

9) Appendix Tables S1-S6 are actually datasets that should be uploaded as one .xsl file per table and renamed to Dataset EV1-X. Each dataset will need its legend removed from the Appendix and added to the corresponding file in a separate tab. Please also be sure to update the source file names, the titles in the submission system and in the files themselves (in the legend), as well as their callouts in the main manuscript text. These tables should then also be removed from the Table of Contents in the Appendix file.

We have renamed these files and amended the text throughout.

10) Synopsis:

- Synopsis image: Please provide a graphic that summarises the main findings of the manuscript on a glance and upload it as a high-resolution jpeg file 550 pixels wide x (300-600) pixels high.

- Synopsis text: Please provide a short standfirst (maximum of 300 characters, including space), limit the bullet points to max. 5 and upload it as a separate .doc file. Please write the bullet points to summarise the key NEW findings. They should be designed to be complementary to the abstract - i.e. not repeat the same text. We encourage inclusion of key acronyms and quantitative information (maximum of 30 words / bullet point). Please use the passive voice.

We provide these files in the submission system.

11) As part of the EMBO Publications transparent editorial process initiative (see our policy here: https://www.embopress.org/transparent-process#Review_Process), Molecular Systems Biology will publish online a Peer Review File (PRF) to accompany accepted manuscripts. This file will be published in conjunction with your paper and will include the anonymous referee reports, your point-by-point response and all pertinent correspondence relating to the manuscript. Let us know whether you agree with the publication of the PRF and as here, if you want to remove or not any figures from it prior to publication. Please note that the Authors checklist will be published at the end of the PRF.

We are happy for a full Peer Review File to accompany our manuscript.

Reviewers Comments:

Reviewer #2: The authors have adequately addressed my questions and concerns. I have no further questions or suggestions for modifications.

We thank the reviewer for their valuable critique and suggestions during the review process.

Reviewer #3: I sincerely thank the authors for their major revision of the manuscript and have no further comments. I recommend publication.

We thank the reviewer for their valuable critique and suggestions during the review process.

Reviewer #4: The authors have addressed many of my concerns and added justifications and explanations to the text. My remaining criticisms are with the inaccurate annotations of certain orders on the website. It is misleading and people that use this website as a resource may not realize that the illustration is not displaying the gene organization.

In the Ebola virus Genome, the L protein is displayed as 3 pieces that look like 3 separate ORFs with 1 in an overlapping reading frame.

This would be a reasonable assumption since it looks the same as how the overlapping reading frames in the influenza A virus Genome are accurately depicted.

The color coding is very helpful, but the complete gene needs to always be displayed. I would be satisfied if every time a protein was split into multiple fragments (displayed in cyan), that the complete gene is shown above in grey, similar to what is done for mature proteins like Hantaan virus G1 and G2.

First, thank you to the reviewer for their detailed and valuable feedback, addressing these and other concerns will allow us to build a maximally useful resource for virology.

This specific comment pertains to the challenges in systematic processing of long protein sequences (and their annotations) for structure prediction. To manage computational demands, we had to make pragmatic choices about how to handle long sequences. We chose to use nested annotations (e.g. protein region, or mature peptide) for sequences >2000 residues. This is comparable to the strategy taken by other high-scale structure prediction efforts. For example, in the EBI AlphaFold Database sequences of length ≥ 2700 residues are broken into arbitrary 1400 residue blocks, whereas in the BFVD, sequences of length ≥ 1500 are similarly fragmented.

Our strategy works well for many polyproteins (e.g. dengue virus 1), but has an unintended consequence of breaking up large (non-polyprotein) proteins, such as Ebola L protein. Clearly, there is not a simple "one-size fits all" solution to these challenges. However, as we prepare for updates of Viro3D we will implement a mixture of strategies including full-length predictions for critical non-polyprotein sequences and improved visual indicators to distinguish fragmented proteins.

Related to this, we anticipated that annotations and sequence handling would be imperfect and, for this very reason, prioritised having clear visual representations of the genomic context of sequences/structures. We would argue that this is a strength of Viro3D, compared to alternative resources. This level of data transparency allows users to better understand what is (and isn't) included for a given virus. Nonetheless, we agree that improvements can be made and are committed to achieving these in the future.

Minor

386: "Gb" should be "gB"

This has been corrected throughout, including in Fig. EV4.

"influenza A virus Genome" is not capitalized but "Ebola" and "Hantaan" are. The capitalization should be consistent.

In Hantaan virus, G1 and G2 are labelled with "see comment". However there is no "comment" on the page.

Thank you for catching these inconsistencies. They both arise from systematically imported metadata from the ICTV VMR and Genbank. As we prepare for future systematic updates we are employing strategies to sanitise and harmonise metadata imports.

1st Sep 2025

Manuscript number: MSB-2025-13039RR

Title: Viro3D: a comprehensive database of virus protein structure predictions

Dear Dr. Grove,

Thank you again for sending us your revised manuscript. I am pleased to inform you that your paper has been accepted for publication in Molecular Systems Biology.

Yours sincerely,

Poonam Bheda, PhD
Scientific Editor
Molecular Systems Biology
